# Comparison of Deterministic and Statistical Models for Water Quality Compliance Forecasting in the San Joaquin River Basin, California

**Nigel W. T. Quinn [1,2,3,*], Michael K. Tansey [2] and Tansey James Lu [2]**

[1] Lawrence Berkeley National Laboratory, Climate and Ecosystem Sciences Division, Berkeley, CA 94720, USA

[2] US Bureau of Reclamation, Washington, DC 20240-0001, USA; mtansey@usbr.gov (M.K.T.); jlu@usbr.gov (T.J.L.)

[3] School of Engineering, University of California, Merced, CA 90095, USA

\* Correspondence: nwquinn@lbl.gov; Tel.: +510-612-8802

**Abstract:** Model selection for water quality forecasting depends on many factors including analyst expertise and cost, stakeholder involvement and expected performance. Water quality forecasting in arid river basins is especially challenging given the importance of protecting beneficial uses in these environments and the livelihood of agricultural communities. In the agriculture-dominated San Joaquin River Basin of California, real-time salinity management (RTSM) is a state-sanctioned program that helps to maximize allowable salt export while protecting existing basin beneficial uses of water supply. The RTSM strategy supplants the federal total maximum daily load (TMDL) approach that could impose fines associated with exceedances of monthly and annual salt load allocations of up to $1 million per year based on average year hydrology and salt load export limits. The essential components of the current program include the establishment of telemetered sensor networks, a web-based information system for sharing data, a basin-scale salt load assimilative capacity forecasting model and institutional entities tasked with performing weekly forecasts of river salt assimilative capacity and scheduling west-side drainage export of salt loads. Web-based information portals have been developed to share model input data and salt assimilative capacity forecasts together with increasing stakeholder awareness and involvement in water quality resource management activities in the river basin. Two modeling approaches have been developed simultaneously. The first relies on a statistical analysis of the relationship between flow and salt concentration at three compliance monitoring sites and the use of these regression relationships for forecasting. The second salt load forecasting approach is a customized application of the Watershed Analysis Risk Management Framework (WARMF), a watershed water quality simulation model that has been configured to estimate daily river salt assimilative capacity and to provide decision support for real-time salinity management at the watershed level. Analysis of the results from both model-based forecasting approaches over a period of five years shows that the regression-based forecasting model, run daily Monday to Friday each week, provided marginally better performance. However, the regression-based forecasting model assumes the same general relationship between flow and salinity which breaks down during extreme weather events such as droughts when water allocation cutbacks among stakeholders are not evenly distributed across the basin. A recent test case shows the utility of both models in dealing with an exceedance event at one compliance monitoring site recently introduced in 2020.

**Keywords:** water quality forecasting; decision support; WARMF; regression model; salinity; irrigated agriculture; stakeholder involvement

## 1. Introduction

Water quality forecasting in arid river basins is especially challenging given the importance of protecting beneficial uses in these environments and the livelihood of agricultural communities. Model selection for water quality forecasting depends on many factors including analyst expertise and cost, stakeholder involvement and expected performance. An American Society of Civil Engineers (ASCE) Task Committee was convened within the Environmental Water Research Institute (ASCE, 2021) to document the state of the practice in the use of water quality models that addresses selection, data collection and organization, calibration, and independent testing to define uncertainty and to envisage both the state of the art and future development. This paper draws on this effort focusing specifically on two distinctly different approaches to involving stakeholders in salinity management in a highly regulated river basin in California, dominated by agricultural and managed wetland return flows.

Management of salinity in the United States and around the world is typically performed through environmental regulation. In the United States, the federal Environmental Protection Agency (USEPA) uses the concept of a total maximum daily load (TMDL) to establish safe and sustainable pollutant concentrations in receiving waters and pollutant load assimilative capacity to help guide stakeholders in determining pollutant load reduction strategies. The TMDL goal for salt loading to impaired waterbody [1,2] can be defined as:

$$TMDL = \sum WLA + \sum LA + MOS + RC \tag{1}$$

where $WLA$ is the waste load allocation for each point source of salt load, $LA$ is the salt load allocation for non-point sources and $MOS$ is the margin of safety selected that accounts for measurement and analytical uncertainty. The $RC$ is a reserve capacity that is seldom used in California applications but that could be used to account for future anticipated loading from both point sources and non-point sources. Possible examples are future climate change, population growth, land use and land cover changes, sea-level rise and environmental policy initiatives.

Models are commonly used in the development of TMDLs and to assess their impact under a range of environmental conditions [2–8]. Models can range in complexity from simple salinity mass balances, that may use simple regression equations to relate salinity to flow and other water quality parameters, to comprehensive, physically-based hydrologic and water quality models [3,9] that attempt to simulate important processes. These models may be used at various phases of TMDL development and implementation including (a) the assessment of the level of impairment and the impacts of existing best management practices on the water quality; (b) the evaluation and comparison of load reduction strategies; (c) the computation of TMDL uncertainty and margin of safety [10]; (d) as decision support tools [11–13]; and (e) for real-time or near-real-time forecasting after implementing a TMDL [14–16]. This paper compares the performance of two modeling techniques used in near-real-time forecasting of compliance with salinity objectives in the San Joaquin River Basin in California.

## 2. Background

The San Joaquin River (SJR) drains approximately 8.7 million acres (4 million ha) of California's San Joaquin Valley including 1.4 million acres (0.64 million ha) of agricultural land (Figure 1). The San Joaquin River Basin (SJRB) watershed is bounded by the Sierra Nevada mountains on the east, the Coast Range mountains on the west, the Sacramento–San Joaquin Delta to the north, and the closed Tulare Lake Basin on the south. The Coast Range mountains are relatively recent in geologic history and formed of an uplifted seabed whose sedimentary constitution is naturally high in salinity including trace elements such as selenium, boron and molybdenum. [17–19]. Additional salt is imported to the basin from large state and federal water pumping facilities in the Sacramento–San Joaquin

Delta. These facilities replace water supply that was diverted from the SJR to irrigate farmland in the southern part of the San Joaquin Valley in the 1960s [17] and constitute more than 47% of the salts imported to the basin. For this reason, as the main purveyor of irrigation water supply, the federal government is considered a stakeholder in actions to manage salinity impairments in the SJR.

Since the 1940s, mean annual salinity concentrations in the SJR measured at the Vernalis monitoring station, the most downstream station not impacted by tidal flows in the Delta, have more than doubled. West-side SJRB sources that include agricultural surface and subsurface drainage and surface drainage from seasonally managed wetlands that comprise the 140,000 acres (64,000 ha) Grasslands Ecological Area discharge through Mud and Salt Sloughs and accounted for more than 37% of the salt loading to the SJR for the period 2000–2009 (Figure 1). Several smaller, ephemeral streams including Hospital, Ingram, Del Puerto, Orestimba and Los Banos Creeks contribute an additional 30% to SJR salt loads [13]. The major tributaries to the SJR, the Stanislaus, Tuolumne, and Merced Rivers, drain the east side of the basin and are the major source of dilution flow and salt load assimilative capacity to the SJR (Figure 1).

Water quality data collected by the Central Valley Regional Water Quality Control Board (CVRWQCB) staff since 1985 indicate that the 30 day running average electrical conductivity (EC) water quality objectives of 1000 μS/cm in the non-irrigation season and 700 μS/cm in the irrigation season (April 1–August 31) have been routinely exceeded at the Vernalis compliance monitoring station, especially prior to 2005 [12,13]. The non-irrigation season salinity objective was exceeded 11 percent of the time and the irrigation season salinity objective was exceeded 49 percent of the time during the period 1986–1998 [13]. This rate of exceedance occurred even though releases were made from New Melones Reservoir on the Stanislaus River to help meet salinity objectives at Vernalis [12].

The SJR TMDL for salinity had several objectives, namely (a) to identify and quantify the sources of salt loading to the SJR; (b) determine the load reductions necessary to achieve attainment of applicable water quality objectives in order to protect beneficial uses of SJR water supply; and (c) to allocate salt loads to the various sources and source areas within the watershed which, once implemented, would result in attainment of applicable water quality objectives [13,14]. Figure 1 shows the seven source areas identified by the CVRWQCB that each were assigned annual and monthly salinity load objectives, modified to account for wet, normal, dry and critically dry water year classifications. However. realization of these objectives using a 10% low flow hydrology to account for critically low-flow conditions over a 73 year historical flow record, in lieu of the standard MOS, produced a TMDL where the base load allocations were overly conservative.

The TMDL already recognized a consumptive use allocation to account for irrigation evapotranspiration of applied water, a Delta Mendota Canal supply relaxation load for salt imported with water supply deliveries to the west side of the basin, a SJR supply water relaxation for salts diverted from the SJR and an allocation to the federal agency for actions related to mitigation of salts imported by the agency in irrigation water supply. The USBR was assigned responsibility for 47 percent of the salt load discharged to the SJR [13].

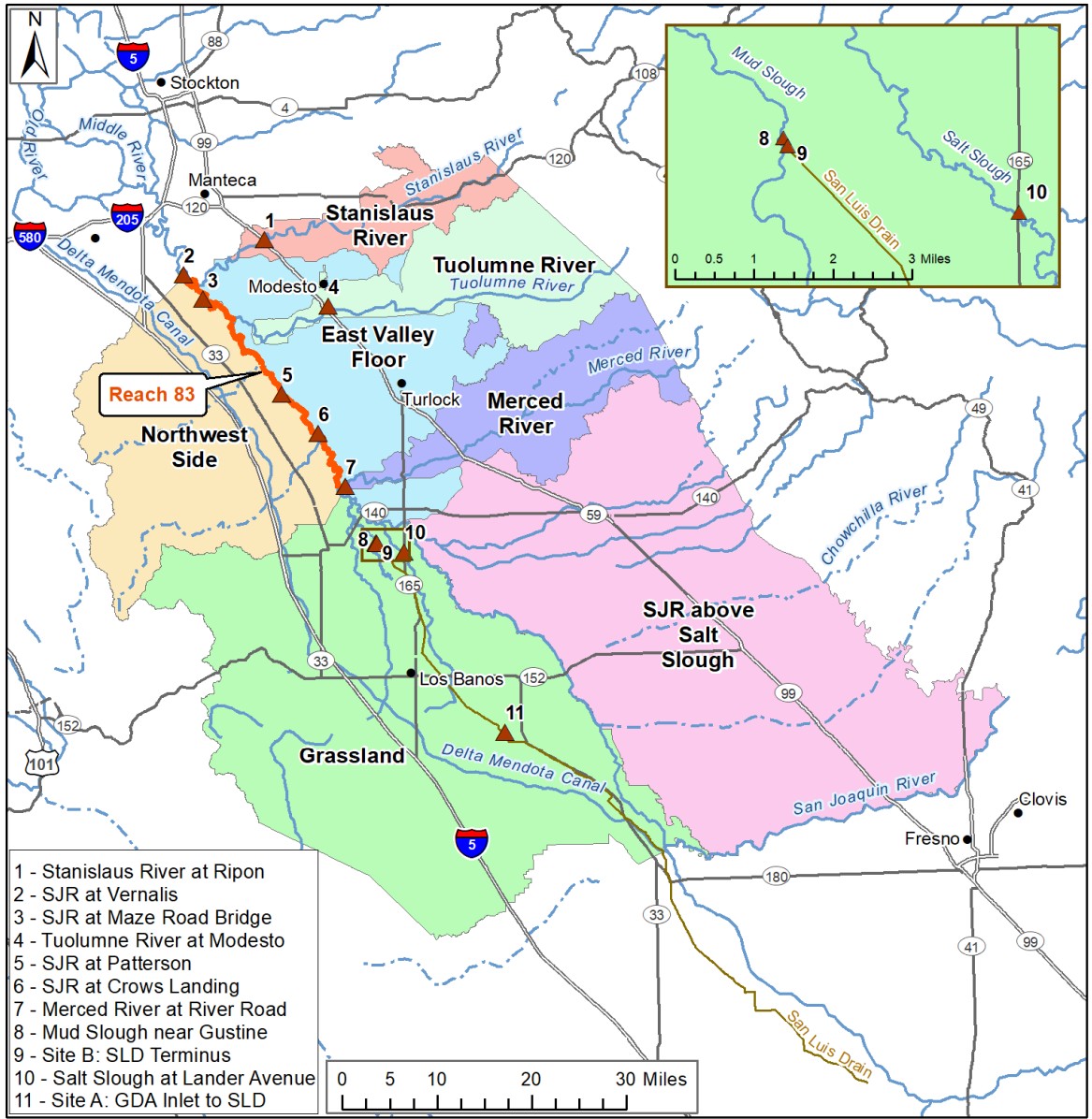

**Figure 1.** Major subareas within the SJR Basin that drain to the SJR as defined in the salinity TMDL [13]. Reach 83 shown in the figure is the reach for which water quality (salinity) is regulated through the recognition of three compliance monitoring stations at Crows Landing, Maze Road Bridge and Vernalis. The most salient feature of the SJR Basin is that drainage from sources to the west of the SJR are elevated in salinity by virtue of native salts in alluvial sediments deposited from the coastal range mountains west of the Valley floor and the importation of irrigation water supply from the Sacramento-San Joaquin Delta that is also salt impacted. Tributary inflow from land areas to the east of the SJR are of high quality, derived from snowmelt from the Sierra Nevada mountains. Real-time management is essentially a scheduling activity—coordinating salt load assimilative capacity consumed by west-side saline drainage with salt load assimilative capacity supplied by east-side reservoir releases along the major tributaries.

Analysis conducted by CVRWQCB staff showed that stakeholder adherence to these salt load limits would result in salt accumulation in the watershed and long term of both degradation of ground and surface waters. Continuation of existing drainage practices could result in average annual fines of over $300,000 in each of the subareas (Table 1) assuming a fine schedule of $5000 per day for each month the load allocation for each subarea was exceeded [20]. These fines would be borne largely by agricultural water district stakeholders, some of whom are those adversely impacted by elevated EC in the SJR

primarily along Reach 83 (Figure 1). Agriculture is the primary beneficial use impaired by salinity in the SJR Basin recognized in the SJR Basin plan [13,14]. To overcome the constraints imposed by the conservative nature of the strict TMDL formulation, the CVRWQCB made provision for an additional real-time salt load allocation in lieu of the fixed base load allocation to maximize salt export from the LSJR Basin while still meeting water quality objectives [14]. The real-time load allocation would apply any time salt load assimilative capacity was available in the SJR.

**Table 1.** Hypothetical SJR daily salt discharge exceedance fees by subarea (10-year period 2001-2012) using an assumed $5000/day fine for exceedance of the 30-day running average mean EC objective [20].

| LSJR Salt Discharge Exceedence Fees by TMDL Subarea for a 10 Year Period 2001–2012 | | | | | |
|---|---|---|---|---|---|
| | | Northwest Side | Grasslands | Upstream San Joaquin River | East Valley Floor |
| | Oct | 0 | 0 | 0 | 0 |
| | Nov | 90 | 60 | 0 | 0 |
| | Dec | 124 | 248 | 0 | 0 |
| | Jan | 186 | 0 | 310 | 0 |
| | Feb | 28 | 196 | 0 | 0 |
| days exceeded by period | Mar | 0 | 279 | 0 | 0 |
| | Apr | 28 | 56 | 42 | 14 |
| | VAMP | 0 | 0 | 30 | 30 |
| | May | 0 | 0 | 51 | 17 |
| | Jun | 30 | 30 | 210 | 90 |
| | Jul | 0 | 0 | 248 | 91 |
| | Aug | 0 | 0 | 248 | 31 |
| | Sep | 0 | 0 | 0 | 0 |
| | Total days of exceedences | 486 | 869 | 1139 | 273 |
| | $5000 per day penalty | $5000 | $5000 | $5000 | $5000 |
| | Total penalties | $2,430,000 | $4,345,000 | $5,695,000 | $1,365,000 |
| | Years calculated | 8 | 10 | 10 | 3 |
| | Average penalty per year | $303,750 | $434,500 | $569,500 | $455,000 |
| | Acres of agriculture | 118,000 | 353,000 | 187,000 | 201,000 |
| | Average penalty per acre | $2.57 | $1.23 | $3.05 | $2.26 |

## 3. Real-Time Salinity Management

The USBR provides water to west-side agricultural and wetland resource contractors via the Delta-Mendota Canal (DMC). The USBR's water rights under which the USBR delivers water to the SJR Basin were amended to require that the USBR meet the 1995 Bay Delta Plan Salinity objectives at Vernalis, which are equivalent to the numeric targets established by the salinity TMDL [13]. Upstream salinity objectives at the Crows Landing Bridge compliance monitoring site was ratified in 2017 to protect riparian diverters downstream of Crows Landing and upstream of the Vernalis compliance monitoring site [21]. The control program requires the USBR to meet DMC salt load allocations or provide dilution flows to create additional assimilative capacity for salt in the LSJR equivalent to DMC salt loads in excess of their allocation. Thec program includes an innovative provision that provides relief from the restrictive salinity load restrictions imposed by the salinity TMDL and codified in the Basin Water Quality Control Plan. This provision states that "Participation in a Regional Board approved real-time management program (RTMP) and attainment of salinity and boron water quality objectives will constitute compliance with this control program" [21]. Participation in the RTMP was designed to promote cooperation and data sharing between entities, effectively replacing a costly salt load-based

regulatory program with a more cost-effective, stakeholder-driven program that permitted full use of the river's assimilative capacity for salt [14,15,21]. Participation in the RTMP also included the development and use of a water quality forecasting model to provide stakeholder decision support and allow stakeholders sufficient time to address anticipated violations of the 30 day running average EC at compliance monitoring stations along the SJR [14]. The WARMF model was chosen for this task [22–24]. Compliance became the collective responsibility of SJRB stakeholders including the USBR.

The RTMP strategy increases potential management flexibility for agricultural, wetland and municipal dischargers to the SJR and provides an opportunity to maximize salt load export from the basin without exceeding environmental objectives. However, it assumes a level of coordination and cooperation amongst stakeholders that does not currently exist. The core elements of this program have led to: (a) the development of a basin-scale, sensor network to collect real-time monitoring of flow and salinity data; (b) an information dissemination system for effective sharing of data among basin stakeholders; (c) a need for continual calibration of the WARMF hydrology and salinity model of in the SJR and its contributing watersheds to improve the accuracy of forecasting and daily assessment of river assimilative capacity; (d) the creation and funding of stakeholder institutional entities responsible for coordinating salinity management actions and ensuring compliance with SJR salinity objectives; and € continued oversight and sanction of the CVRWQCB [14–16].

### 3.1. WARMF Water Quality Simulation Model

The San Joaquin River Basin application of the public-domain, Watershed Analysis Risk Management Framework (WARMF) model [14,24] was developed in 2004 by Systech Water Resources Inc. as a TMDL decision support tool. The first application of the model was to assess options for control of dissolved oxygen sag in the SJR Deep Water Ship Channel [23–25]. The SJRB WARMF model application is a physically-based, data-intensive watershed model that simulates the hydrologic, chemical, and physical processes in the river and contributing waterbodies (Figure 2). The model was derived from the San Joaquin River Input–Output (SJRIO) model [26,27]. The model was updated and reconfigured as a salinity forecasting tool in 2014 [14,24] as the USBR's contribution to stakeholder-led real-time salinity management activities. The WARMF model application simulates flow and water quality in surface water diversions, groundwater pumping, and irrigation water supply, while keeping track of crop evapotranspiration, seepage, and irrigation surface and subsurface return flows [25]. Delineation of land catchments in WARMF conforms to both irrigation and drainage district boundaries and natural catchments, allowing the model to track salt loads from their points of diversion in delivery canals back to the river [25].

The data-intensive WARMF model is supplied with daily meteorology, diversion flows, and measured flow and electric conductivity (EC) at the upstream model boundaries [23,25]. The current upstream model boundaries are at gages where flow and EC are measured continuously in the SJR and along its major tributaries including the Merced River, Tuolumne River and Stanislaus River. Real-time data, tributary reservoir release forecasts, and meteorology forecasts are collected and imported into WARMF using an automated process consisting of custom scripts and web scraping tools that interact with agency web portals for hydrology and water quality monitoring [25,28]. WARMF model data acquisition accesses seven agency web portals and is accomplished as a separate data acquisition and pre-processing routine

The combination of real-time monitoring, simulation modeling and forecasting of SJR assimilative capacity has the potential to optimize use of available river salt assimilative capacity, generated by releases of high quality Sierran water, which provides dilution to saline west-side agricultural and managed wetland return flows. However, there needs to be coordination and sufficient lead time to allow entities being asked to charge drainage practices or alter reservoir release patterns to be able to respond. Agricultural return flows

and salt loads are highest during the summer irrigation season whereas return flows and salt loads from seasonally managed wetlands are highest during the spring months of March and April, when most seasonal wetland ponds are drained to promote establishment of moist soil plants and habitat for waterfowl [15]. These anticipated hydrologic patterns help to screen the array of practices on both the east and west sides of the basin that will be most effective at managing salinity.

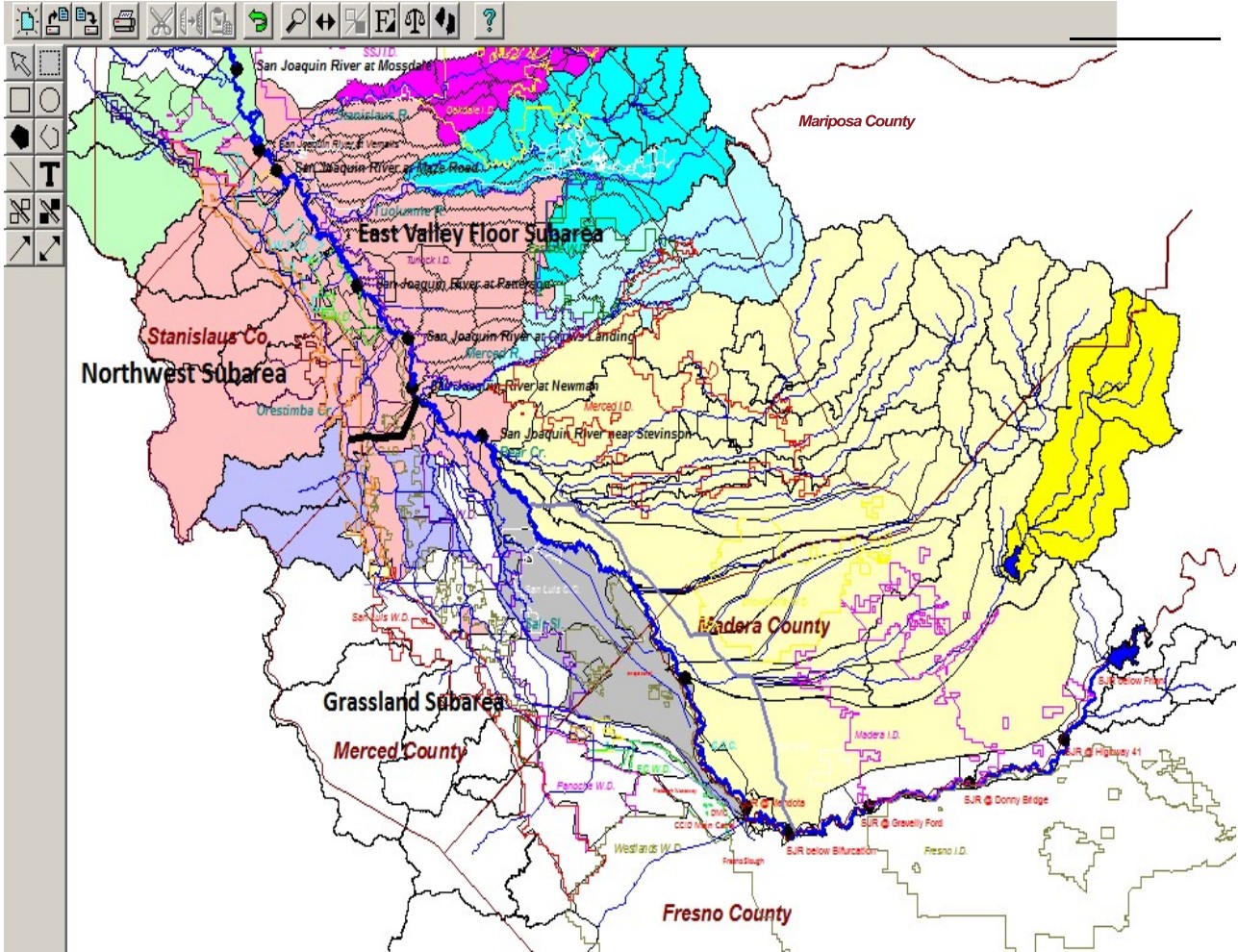

**Figure 2.** Map of the SJR Basin represented as major contributing watersheds within the WARMF model. The WARMF model allows further disaggregation of these watersheds into small contributing subareas and allows the substitution of available data at the major outlets of these subareas for model-derived flow and water quality estimates.

Given the uncertainty associated with estimates of salt assimilative capacity, the need for adequate lead time for stakeholders to adjust tributary inflow and drainage return flow schedules and the fact that most weather forecasts provided by news organizations rarely extend beyond two weeks—a two-week forecast period and a one-week hindcast period was chosen for the real-time salinity management program. The one-week hindcast refers to the technique of beginning the simulation one week in arrears so that the first week of the forecast can be compared to observed flow and electrical conductivity (EC) data [14,16,22]. Model parameters affecting river and tributary inflow and water quality such as the partitioning coefficients that allocate watershed runoff and deep percolation to groundwater can be adjusted to recalibrate the model during periods when model output and river observations diverge. This activity is infrequently performed due to the significant effort involved and the fact that the WARMF model has exhibited

excellent performance for simulation of flow EC and EC along Reach 83 of the SJR. Simulated flow and EC are compared to measured data along the SJR for model calibration including drainage return flows from east- and west-side catchments and direct diversions from the SJR to riparian water districts. Although agricultural and managed wetland stakeholders have yet to fully embrace the model as a decision support tool both have concurred that the suggested two-week forecast and one-week hindcast periods are a good compromise balancing the utility and credibility of the forecasts with the time stakeholders might need to adjust water management and drainage discharge operations.

The SJR WARMF model has a number of customized output visualization options designed to enhance user understanding of salinity fate and transport in the SJR Basin and the use of salt load assimilative capacity by river mile along the mainstem of the SJR [28]. The output visualization also allows users to estimate if and when the salinity concentration at the compliance monitoring sites will approach or exceed objectives. The model is also capable of showing the impact of potential salinity management changes in the watershed designed to comply with regulatory limits (Figure 3). For example, the SJR WARMF model can simulate the effect of increased irrigation water diversions from the river into riparian water districts, lowering salt loading in the river, which may help to improve compliance with salinity concentration objectives [28].

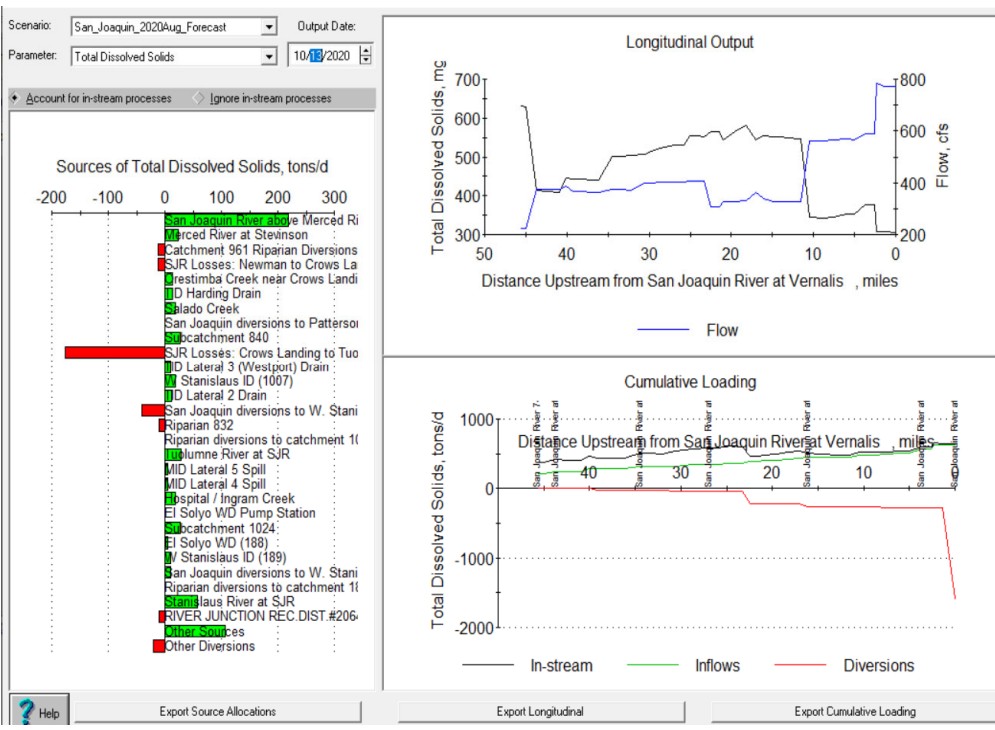

**Figure 3.** A unique feature of the WARMF model is the availability of customized model outputs such as the "Gowdy" output (named after its developer) shown here. This depicts a Lagrangian view of the SJR at any point in time showing the major inflow to and diversions from the river approximately every ½ mile (800 m) along its main reach as well as the incremental flow and EC concentration from the origin at Lander Avenue to the EC compliance monitoring station at Vernalis [23,25].

The SJR WARMF model has been improved and customized over the past 15 years with the USBR and research grant support as a watershed-based simulation tool for flow and salinity forecasting in the SJR [14,25]. Updating time series data inputs and maintaining model calibration are expensive and time consuming. This constraint has restricted the stakeholders' pool and agency individuals able to run the model on a regular basis and has been an impediment for stakeholder entities such as the San Joaquin Valley Drainage Authority (SJDVA) to take over operation and maintenance of the model as a decision

support tool. As a result, the USBR evaluated other approaches for providing flow and salinity forecasts of SJR at Vernalis and Crows Landing, the two salinity concentration compliance points for the TMDL. Although the WARMF model has been used for various decision support activities in the SJR for over 15 years, other less data-intensive and more easily understood approaches may be better received by stakeholders [28].

### 3.2. ANN-Based Statistical Models

The USBR developed a statistical approach as an alternative to the physically-based SJR WARMF model for flow and salinity forecasting in the SJR. This approach was limited to the Vernalis, Crows Landing and Maze Road Bridge compliance monitoring stations (Figure 1) [29]. Two Artificial Neural Networks (ANN) based models a Recurrent ANN and an Autoregressive ANN were identified as potential alternatives [30]. The most salient features of these ANN alternatives were that the underlying basis should be easy to understand and independent of having a deep understanding of basin hydrology [29,30]. ANN and regression-based approaches have the advantage of ready automation and have the advantage that daily flow forecasts are available online from the National Oceanic and Atmospheric Administration (NOAA) California River Forecast Center (RFC) providing the basis for river EC forecasts at the compliance monitoring stations. The significance of this work is that daily bulletins from dam operators along the three major tributaries to the SJR are recognized in these forecasts.

Under normal basin hydrologic conditions, there is sufficient salt load assimilative capacity in the river when defined as the 30 day running average EC. Only in rare circumstances such as a prolonged drought is action required to limit salt loading during certain months to the SJR. During these periods, the more comprehensive WARMF model could be called upon to assist stakeholder management entities to determine appropriate salt loading reduction by subarea within the basin to avoid fines.

Recurrent ANN models are statistical learning models that are used in machine learning, inspired by biological neural networks such as in the human brain [30]. A number of ANN and recurrent neural network architectures with both short- and long-term memory were developed and applied to the Vernalis compliance monitoring station using existing flow and salinity data resources. None of the ANN architectures or network hyper-parameters performed sufficiently well due to time series water quality data limitations and the impact of random anthropogenic factors that can affect reservoir operations [29]. In conducting the analysis, less than 5000 observations were available, whereas most applications of this method typically require well over a million observations to be successful. An additional ANN-based model was investigated using the MATLAB machine learning toolbox using an embedded machine learning application called Autoregressive ANN that accommodated external inputs. Although the Autoregressive ANN approach performed better in salinity forecasts compared to the Recurrent ANN model, the model salinity forecast performance was unsatisfactory [29]. Future work in the application of neural networks to flow and EC time series forecasting on the SJR may find more success in the use of Bayesian neural networks for capturing water quality forecast uncertainty.

### 3.3. Simple Regression Model

Water agency analysts have long recognized the inverse relationship between flow and EC. This relationship was utilized for many years in applications of the previous USBR water supply allocation models for the federal service area within the San Joaquin Valley to estimate New Melones reservoir releases for water quality. However, the poor performance of these models for estimating EC at low-flow conditions, based on simple regression relationships, was one of the reasons a data-driven flow and salinity mass balance approach was adopted for the state-federal California (Water Allocation) Simulation Model (CALSIM) model that replaced the previous models. A re-examination of the flow–EC relationship [29] suggested a new approach using the rate of change of salinity that was found to be approximately proportional to the rate of change (or gradient) of the

measured flow in the SJR. This new algorithm was not as susceptible to low-flow conditions as the prior approach.

The flow gradient was calculated as follows:

$$Q_{grad} = (Q_t - Q_{(t-1)})/Q_{(t-1)}$$

where $Q_t$ is the flow at time t, and $Q_{(t-1)}$ is the flow at the previous time step.

The salinity gradient was calculated in a similar fashion. Further analysis of daily flow and salinity data of the SJR at Vernalis for the period 2000–2018 showed that a clear linear regression relationship exists between flow and salinity gradients. After removing one percent of the outliers from the plot of flow and salinity gradients using daily data for the 2000–2018 time period, the resulting regression equation of flow and salinity relationship at Vernalis became (Lu et al., 2019):

$$EC_{grad} = -0.5396 \times Q_{grad} + 0.0038$$

or

$$[(EC)_t - [EC_{(t-1)})/ [EC)_{(t-1)}] = -0.5396 \times Q_t - Q_{(t-1)})/Q_{(t-1)} + 0.0038$$

Using this relationship, the salinity forecast (measured as EC) at time step t can be determined as follows:

$$[EC]_t = [EC]_{(t-1)} - [0.5396 \times (Q_t - Q_{(t-1)})/Q_{(t-1)} + 0.0038] \times [EC]_{(t-1)}$$

This equation was initially applied to daily Vernalis flow and salinity data (Figure 4) for the period 2000–2018 to generate six-day model-based forecasts that were compared to historical data. The correlation coefficients for the relationship between the six-day forecasted salinity and observed flow ranged from 0.8780 to 0.9787. The same regression method was then applied to the upstream Crows Landing compliance monitoring station, resulting in the following equation for forecasting the SJR salinity concentration downstream of that location.

$$[EC]_t = [EC]_{(t-1)} + [-0.4413 \times (Q_t - Q_{(t-1)})/Q_{(t-1)} + 0.0036] \times [EC]_{(t-1)}$$

The correlation coefficients of the relationship of observed flow and the six-day forecasted salinity concentration ranged from 0.9831 to 0.9154.

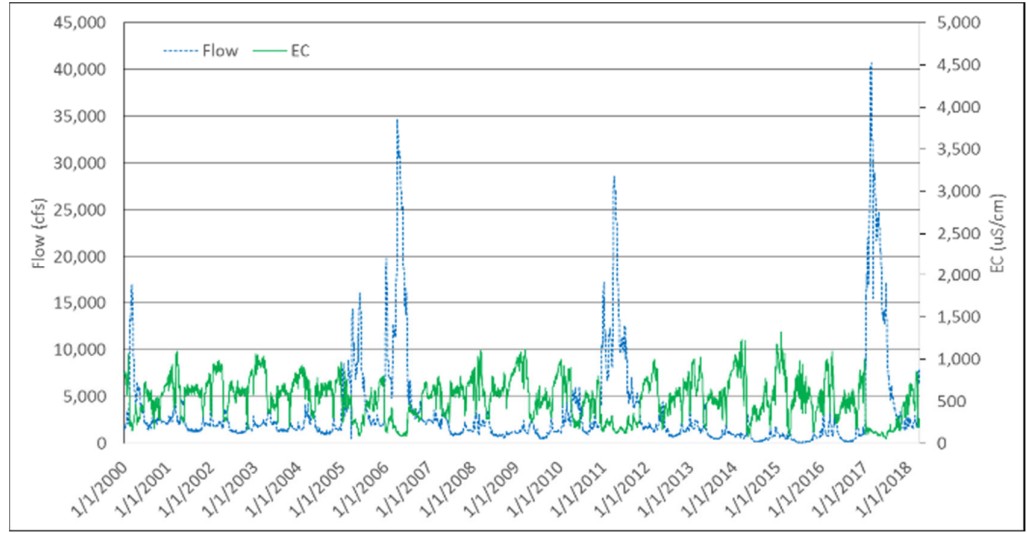

**Figure 4.** Flow and EC observations at Vernalis compliance monitoring station on the SJR for the period 2000–2018.

## 4. Comparison of the SJR WARMF and Regression Model Applications

A comparison of the SJR WARMF and Regression models was undertaken to evaluate the performance of the models for water quality forecasting. This evaluation initially compared differences between forecasted and observed water quality measured as EC at the monitoring station located at Vernalis (Figure 4). Similar analyses were performed in Excel using an algorithm that computed the difference ($\Delta$) between the daily forecasted (FC) and observed (OBS) EC ($\Delta$ = FC – OBS) starting on the forecast day (FC Day + 0) and each consecutive day within the lead forecast time of 14 days (FC Day + 14). The analyses were conducted with observations and forecasts made between 22 February 2018 and 22 May 2020. During this period, a total of 820 EC observations were measured. However, for all the forecast lead times considerably fewer forecasts were actually made. In the case of the Regression model, the number of forecasts ranged from 399 for forecasts of less than 6 days (FC Day + 6) down to 347 forecasts for lead times of 7 days or more (FC + 7 to FC + 14). Forecasts were made only on regular workdays and were not conducted on certain days due to personnel availability and periods of downtime in the monitoring system. Forecasts for days 11 through 15 were simply repeats of the FC + 10 forecast given that the California River Forecast Center (RFC) does not extend its daily forecasts, used by the WARMF and Regression models, past 10 days.

In the case of the WARMF model, there were even fewer forecasts throughout the evaluation period. The greater personnel time commitment to make WARMF model forecasts limited the forecast frequency to once per week, usually on a Monday. There were 131 forecasts for lead times from FC + 0 to FC + 7 and fewer forecasts for greater lead times. Table 2 presents the frequency count and statistics (mean and standard deviation) for the observations and model forecasts in the initial comparison of results produced by the Regression and WARMF models. Table 2 confirms that the Regression model forecasts were made approximately 3 times more often than those for the WARMF model.

In general, the Regression model forecasts had mean EC predictions that are approximately equal to the mean EC of the observations but increased to above the observation's mean EC after FC Day + 5 through the end of the forecast period. The WARMF model had slightly lower mean forecast EC values until FC Day + 4 after which they increased throughout the remainder of the forecast period. The observed EC, Regression and WARMF forecast mean EC values were compared in Figure 5 at each of the forecast lead times.

**Table 2.** Statistics of observed (OBS) and forecasted (FC) EC ($\mu$S/cm) for the Regression and WARMF models made between 22 February 2018 and 22 May 2020 by lead time.

| Regression Model EC Data | | | | WARMF Model EC Data | | | |
|---|---|---|---|---|---|---|---|
| | Count | Mean | Std Dev | | Count | Mean | Std Dev |
| OBS Day + 0 | 399 | 397 | 224 | OBS Day + 0 | 131 | 401 | 235 |
| FC Day + 0 | | 397 | 223 | FC Day + 0 | | 384 | 192 |
| OBS Day + 1 | 399 | 395 | 224 | OBS Day + 1 | 131 | 383 | 214 |
| FC Day +1 | | 397 | 225 | FC Day + 1 | | 381 | 182 |
| OBS Day + 2 | 399 | 393 | 224 | OBS Day + 2 | 131 | 376 | 211 |
| FC Day + 2 | | 394 | 225 | FC Day +2 | | 375 | 178 |
| OBS Day+ 3 | 399 | 393 | 225 | OBS Day + 3 | 131 | 377 | 208 |
| FC Day + 3 | | 393 | 225 | FC Day + 3 | | 374 | 182 |
| OBS Day + 4 | 399 | 394 | 223 | OBS Day + 4 | 131 | 374 | 209 |
| FC Day + 4 | | 393 | 226 | FC Day + 4 | | 372 | 183 |
| OBS Day + 5 | 399 | 393 | 222 | OBS Day + 5 | 131 | 370 | 207 |
| FC Day + 5 | | 394 | 224 | FC Day + 5 | | 375 | 187 |
| OBS Day + 6 | 398 | 391 | 219 | OBS Day + 6 | 131 | 371 | 201 |
| FC Day + 6 | | 395 | 222 | FC Day + 6 | | 380 | 190 |
| OBS Day + 7 | 347 | 394 | 218 | OBS Day + 7 | 131 | 373 | 204 |
| FC Day + 7 | | 400 | 218 | FC Day + 7 | | 387 | 194 |
| OBS Day + 8 | 347 | 393 | 217 | OBS Day + 8 | 129 | 370 | 203 |
| FC Day + 8 | | 402 | 220 | FC Day + 8 | | 390 | 200 |

| | | | | | | | | |
|---|---|---|---|---|---|---|---|---|
| OBS Day + 9 | 347 | 392 | 218 | OBS Day + 9 | 129 | 366 | 202 |
| FC Day + 9 | | 405 | 223 | FC Day + 9 | | 391 | 204 |
| OBS Day + 10 | 347 | 395 | 222 | OBS Day + 10 | 128 | 366 | 204 |
| FC Day + 10 | | 408 | 225 | FC Day + 10 | | 393 | 208 |
| OBS Day + 11 | 347 | 398 | 224 | OBS Day + 11 | 128 | 366 | 207 |
| FC Day + 11 | | 408 | 225 | FC Day + 11 | | 393 | 211 |
| OBS Day + 12 | 347 | 397 | 223 | OBS Day + 12 | 126 | 363 | 203 |
| FC Day + 12 | | 408 | 225 | FC Day + 12 | | 395 | 213 |
| OBS Day + 13 | 347 | 397 | 225 | OBS Day + 13 | 126 | 363 | 204 |
| FC Day + 13 | | 408 | 225 | FC Day + 13 | | 395 | 214 |
| OBS Day + 14 | 347 | 398 | 229 | OBS Day + 14 | 124 | 370 | 209 |
| FC Day + 14 | | 408 | 224 | FC Day + 14 | | 399 | 214 |

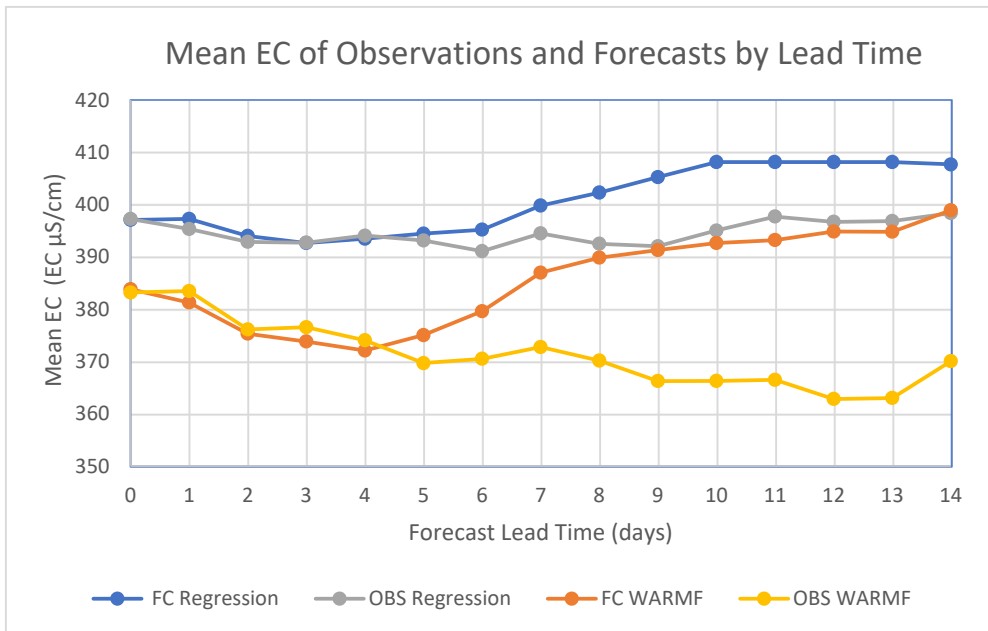

**Figure 5.** Means of the observed (OBS) EC and Forecast (FC) EC for the Regression and WARMF models for all forecast lead times between 22 February 2018 and 22 May 2020.

A comparison of the mean of differences between forecasted EC and observed EC for both Regression and WARMF models is shown in Table 3. For both models, the mean of the differences between forecasted EC minus observed EC was computed for the period between 22 February 2018 and 22 May 2020. For the Regression model, the differences were small (≤+5) for until Δ Day + 6. The mean EC differences increase to maximum of 15 μS/cm at Δ Day + 9. From Δ Day + 10 to the end of the forecast period, the mean EC differences decrease slightly to a value of 12 μS/cm. For the WARMF model, the mean of the EC differences were small, decreasing from +1 to −3 at Δ Day + 3. From Δ Day + 4 to Δ Day + 12, the mean of the EC differences increases consistently reaching a peak value of +33 μS/cm at Δ Day + 12 after which there is a slight decrease to 30 μS/cm at the end of the forecast period. These results are illustrated in Figure 6.

**Table 3.** Comparison of mean differences (Δ) between forecasted EC and observed EC (μS/cm) for all model forecasts made between 22 February 2018 and 22 May 2020.

| | Regression Model EC Differences | | | | WARMF Model EC Differences | | |
|---|---|---|---|---|---|---|---|
| | **Count** | **Mean Δ** | **Std Dev Δ** | | **Count** | **Mean Δ** | **Std Dev Δ** |
| Δ Day + 0 | 398 | 0 | 14 | Δ Day + 0 | 131 | 1 | 78 |
| Δ Day + 1 | 397 | 2 | 37 | Δ Day + 1 | 131 | −2 | 85 |
| Δ Day + 2 | 396 | 2 | 48 | Δ Day + 2 | 131 | −1 | 92 |
| Δ Day + 3 | 395 | 1 | 57 | Δ Day + 3 | 131 | −3 | 86 |

| Δ Day + 4 | 394 | 1 | 69 | Δ Day + 4 | 130 | −2 | 100 |
| Δ Day + 5 | 394 | 3 | 80 | Δ Day + 5 | 130 | 5 | 105 |
| Δ Day + 6 | 393 | 5 | 86 | Δ Day + 6 | 130 | 9 | 108 |
| Δ Day + 7 | 341 | 7 | 91 | Δ Day + 7 | 130 | 14 | 115 |
| Δ Day + 8 | 340 | 12 | 103 | Δ Day + 8 | 128 | 20 | 122 |
| Δ Day + 9 | 339 | 15 | 116 | Δ Day + 9 | 128 | 25 | 134 |
| Δ Day + 10 | 338 | 15 | 131 | Δ Day + 10 | 127 | 27 | 142 |
| Δ Day + 11 | 337 | 13 | 144 | Δ Day + 11 | 126 | 27 | 151 |
| Δ Day + 12 | 337 | 14 | 153 | Δ Day + 12 | 124 | 33 | 164 |
| Δ Day + 13 | 337 | 14 | 163 | Δ Day + 13 | 124 | 33 | 173 |
| Δ Day + 14 | 336 | 12 | 171 | Δ Day + 14 | 122 | 30 | 179 |

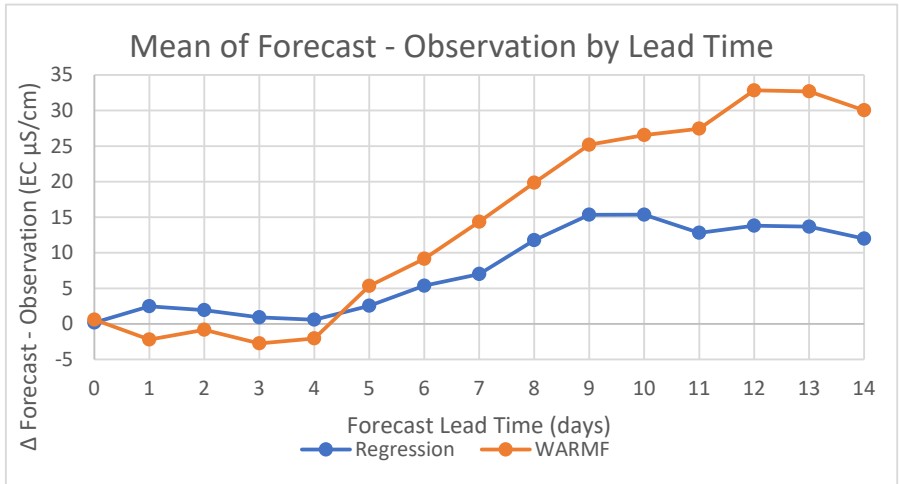

**Figure 6.** Comparison of mean differences in forecasted EC and observed EC for the Regression and WARMF models for the period between 22 February 2018 and 22 May 2020.

The forecast standard deviation is a measure of the dispersion of the forecast EC predictions around the mean EC value. Larger standard deviations imply a wider range of forecast predictions of EC and/or differences between forecasted EC values and observed EC. Figure 7 presents the standard deviations of the EC observations and EC forecasts for both models (Figure 7a) as well as the standard deviations of the EC differences between the forecasts minus observations (Figure 7b) over the forecast period. As illustrated, the standard deviations of the Regression model EC forecasts closely approximate the standard deviations of the EC observations at all lead times. In contrast, the standard deviations of the WARMF model EC forecasts are consistently less than standard deviations of the EC observations until lead time day 8 as shown in Figure 7a. The maximum difference (33 µS/cm) between forecast and observation standard deviations occurs at lead time day 2. In Figure 7b, the standard deviation of the differences between the EC forecasts minus EC observations for both models increase consistently with lead time indicating increasing uncertainty in the EC forecasts. Additionally, the WARMF model has consistently greater standard deviations in EC differences relative to the Regression model.

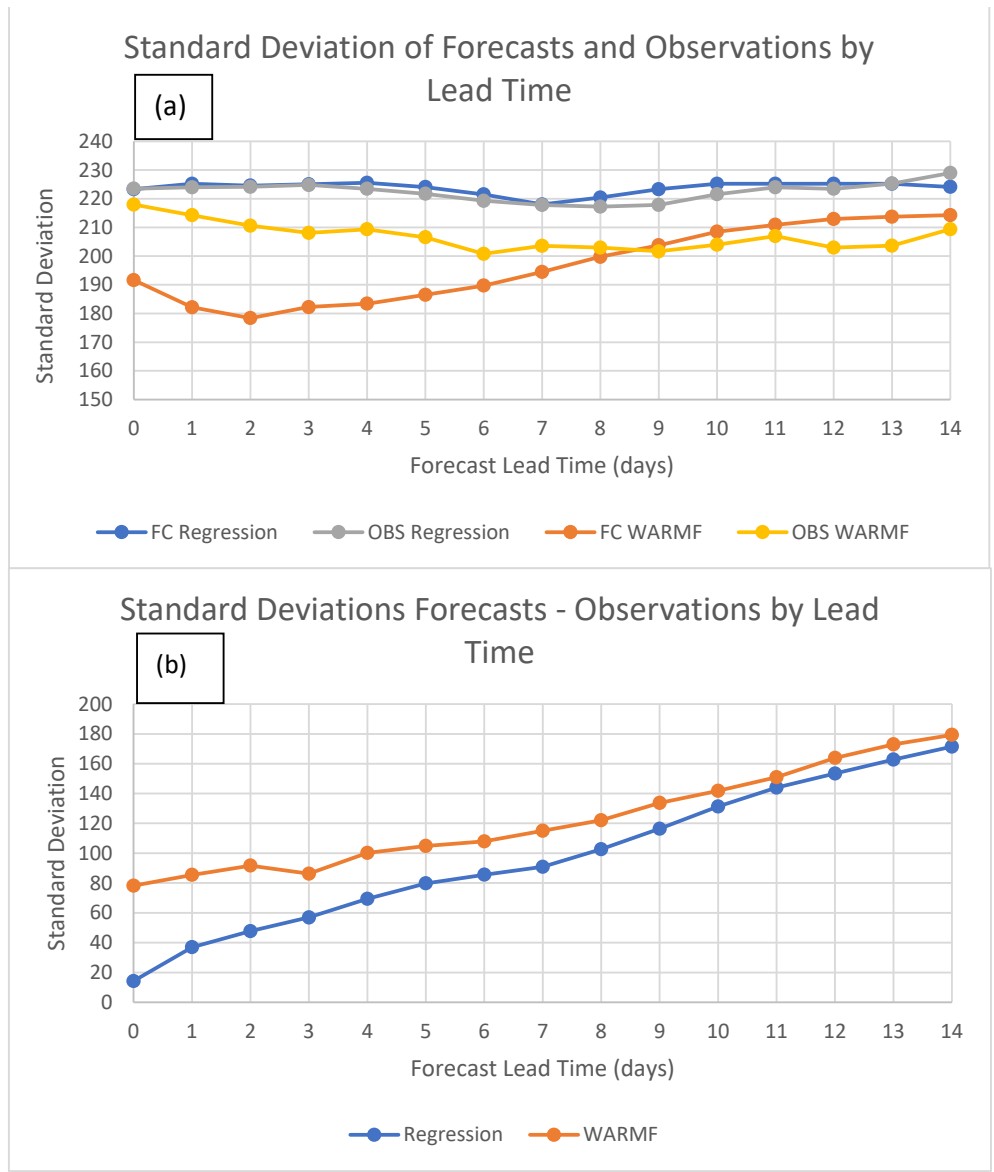

**Figure 7.** (**a**,**b**). Comparison of the standard deviations of forecasted EC and observed EC and standard deviations of differences between EC forecasts and EC observations for the Regression and WARMF models by lead time in the period between 22 February 2018 and 22 May 2020.

An additional evaluation was performed to determine the extent to which model bias affects the mean of differences between the forecasts and observations. For example, the models could forecast values significantly greater than the observations. However, a few large underestimates could potentially offset the positive bias and make the model appear to show better performance. In order to examine this effect, forecasts which were greater than the corresponding observations were examined separately from those in which the forecasts were less than the corresponding observations. After this sorting into positive (forecast >= observation) and negative (forecast < observation) bias groups, the means of the EC differences (forecast–observation) over the study period were calculated for each forecast lead time.

Figures 8 and 9 illustrate comparisons of the Regression and WARMF models for the positive and negative bias results, respectively. For the positive bias differences, the Regression model has lower differences at all lead times than the WARMF model.

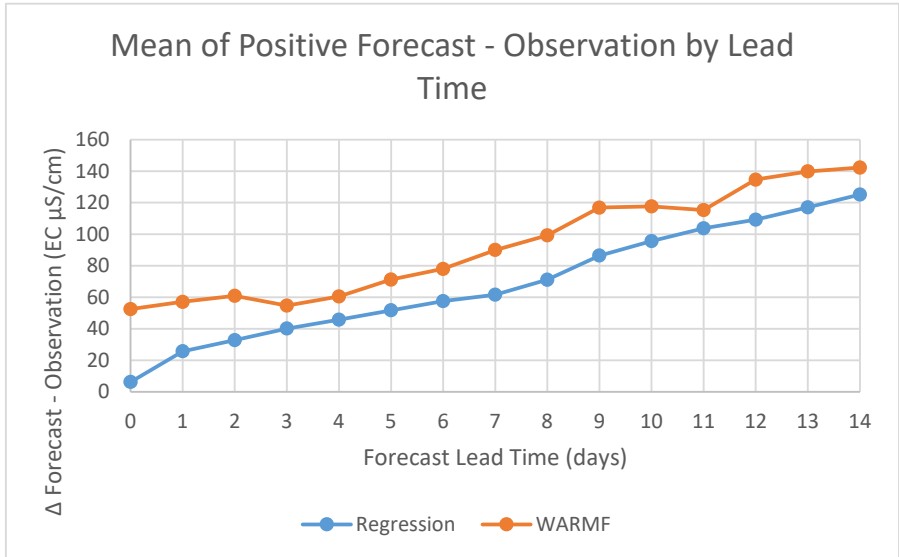

**Figure 8.** Comparison of means of forecasted EC and observed EC for the Regression and WARMF models for the period between 22 February 2018 and 22 May 2020. Data censored to include only over (positive) predictions.

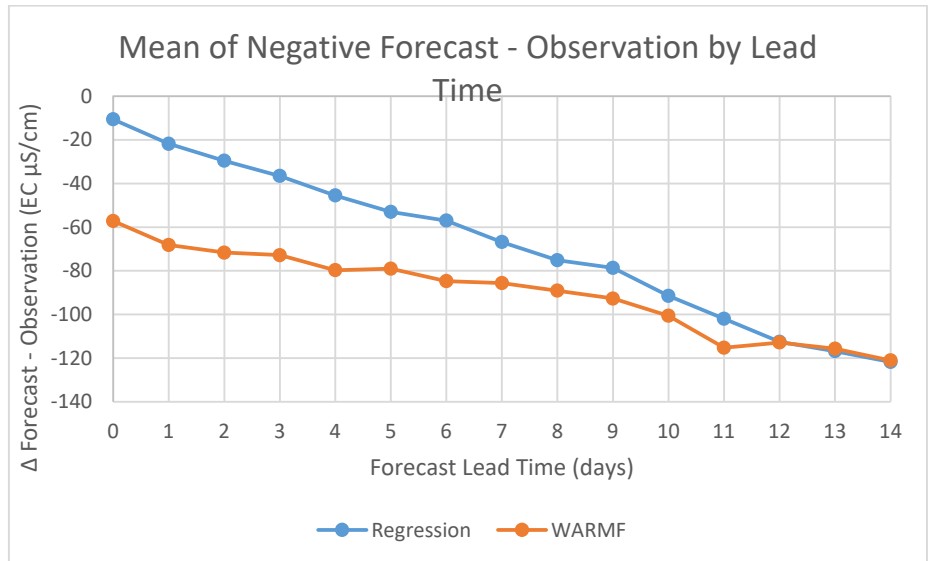

**Figure 9.** Comparison of means of forecasted and observed EC for the Regression and WARMF models for the period between 22 February 2018 and 22 May 2020. Data censored to include only under (negative)-predictions.

For the negative bias differences, the Regression model has lower negative mean differences than the WARMF model from Δ Day + 0 to Δ Day + 11 after which both models have nearly equal EC differences.

Another aspect of the potential bias introduced by these forecasting methods is how frequently do the overpredictions (positive) or underpredictions (negative) of mean differences in EC occur as a function of forecast lead times. For instance, the mean EC forecast bias could be overly influenced by a small number of very large EC discrepancies—either

positive or negative. Figure 10 compares the percentages of positive bias differences in EC for both models.

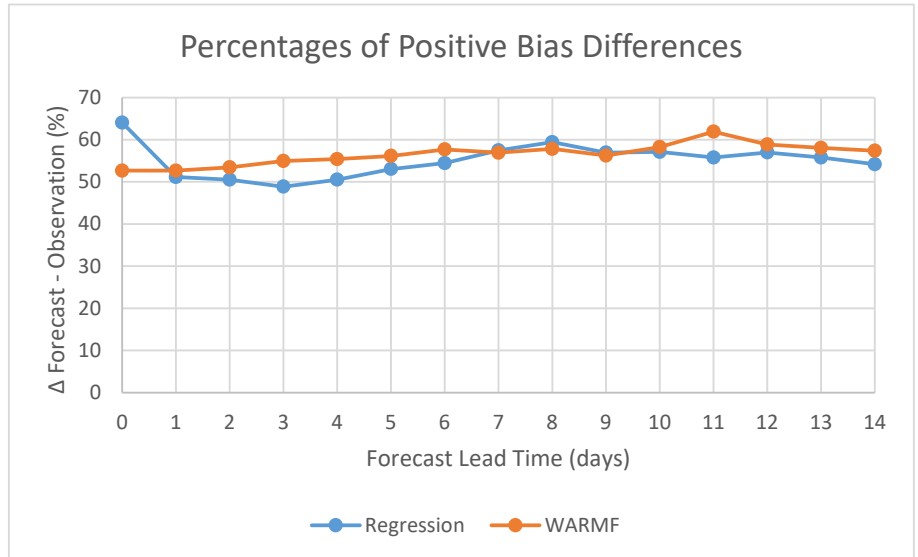

**Figure 10.** Comparison of the percentages of higher (positive bias) EC forecasts for the Regression and WARMF models for the period between 22 February 2018 and 22 May 2020.

As illustrated above, both the Regression and WARMF models exhibit a slight positive EC forecast bias. The Regression model exhibits a higher frequency (65%) of positive EC forecast bias differences on Δ Day + 0 for the period between 22 February 2018 and 22 May 2020. From Δ Day + 1 to Δ Day + 4, the Regression model has a neutral EC forecast bias frequency of approximately 50%. From Δ Day + 5 to Δ Day + 8, the Regression model EC forecast bias becomes increasingly positive reaching a maximum of 60% before declining gradually to 55% by Δ Day + 14. The WARMF model exhibits a gradually increasing positive EC forecast bias from 53% on Δ Day + 0 to 58% on Δ Day + 6. Subsequently, the EC forecast bias declines slightly to Δ Day + 9.

In summary, the results of the model comparison analyses indicate that the Regression model EC forecasts were closer to the overall mean of the EC observations than the WARMF model forecasted EC (Figure 5). As illustrated by Figure 6, the Regression model provided EC forecasts with mean differences of less than or equal to 5 μS/cm for the first 7 days (Δ Day + 0 to Δ Day + 6). In comparison, the WARMF model provided EC forecasts with mean differences of less than or equal to 5 μS/cm for only 5 days (Δ Day + 0 to Δ Day + 4). Based on these measures of performance, the Regression model provided EC forecasts with reduced error relative to the WARMF model especially for the period from Δ Day + 4 to Δ Day + 6.

The standard deviations of Regression model EC forecasts closely approximated the standard deviations EC observations at all lead times. In contrast, the standard deviations of the WARMF model EC forecasts were consistently less than the corresponding standard deviations of the EC observations at lead time less than day  (Figure 7a). For both models, the standard deviation of EC forecast differences steadily increased with forecast lead time, as expected, while the WARMF model had higher standard deviations of EC than the Regression model throughout the forecast period (Figure 7b).

When the EC forecasts were separated into those with overestimate (positive) and underestimate (negative) biases, the mean differences between the EC forecasts and observations were seen to increase predictably with forecast lead times. For both the positive and negative forecast EC mean differences, the Regression model performed better than

the WARMF model for lead times from Δ Day + 0 to Δ Day + 10. From Δ Day + 12 to Δ Day + 14, the performance of both models was approximately the same.

As illustrated in Figure 10, both models have slightly positive EC forecast biases. With the exception of the high overprediction (positive) bias (65%) for the Regression model EC on Δ Day + 0, the Regression model predictions were relatively unbiased between Δ Day + 1 to Δ Day + 4 and subsequently remained slightly positively biased throughout the remainder of the forecast period. The WARMF model made consistently greater overpredictions (positive biases in EC) than the Regression model.

It is also important to note that the Regression model EC and WARMF model EC results were originally based on different forecasted flows. Up until mid-2020, the WARMF model used prior water year operations forecast for the 14 day flow forecast along the three major east-side tributaries. From July 2020 onward, the WARMF model has been using the same flow forecasts as the Regression model which come directly from the NOAA California-Nevada River Forecast Center. The analyst who makes these daily forecasts is in regular communication with reservoir operators at Modesto Irrigation District, Merced Irrigation District and the USBR Central Valley Operations Office who control releases and provide regular bulletins of changes in release schedules. Hence, any differences between the models are no longer a function of the flow release forecasts but rather the WARMF model's watershed simulation and prior knowledge of diversions and drainage inflow along each tributary research and along the mainstem of the SJR.

## 5. Time Series Comparisons of the WARMF and Regression Models

The preceding analysis focused on comparisons of mean EC values and differences between model-predicted EC and observations for the Regression and WARMF models for various forecast lead times. In this section, time series comparisons of the EC predicted by each model compared to EC observations for the same time period were made for selected lead times. As shown in Figure 11, both models have relatively small mean EC differences at forecast lead times of less Δ Day + 4. From Δ Day + 5 to Δ Day + 8, mean differences increased. After Δ Day + 9, the EC predictions of both models reached a relatively constant plateau. Figure 12 also shows a comparison of Regression model EC forecasts and observations at Δ Day + 4, Δ Day + 8 and Δ Day + 12. As illustrated, there was a good match between observations and forecasts. However, as the forecast lead time increased the differences between model forecast of EC and observations also increased. This relationship between model EC forecasts and observations can be quantified using the root mean square error (RMSE) statistic which increases from 69.4 at Δ Day + 4 to 103 at Δ Day + 8 to 154 at Δ Day + 12. Figure 12 shows a similar relationship between model EC forecasts and observations for the WARMF model. In this case, the RMSE increases from 99.8 at Δ Day + 4 to 123 at Δ Day + 8 to 166 at Δ Day + 12. As illustrated by the figures and RMSE values, the Regression model performed somewhat better than the WARMF model in predicting EC for similar lead times.

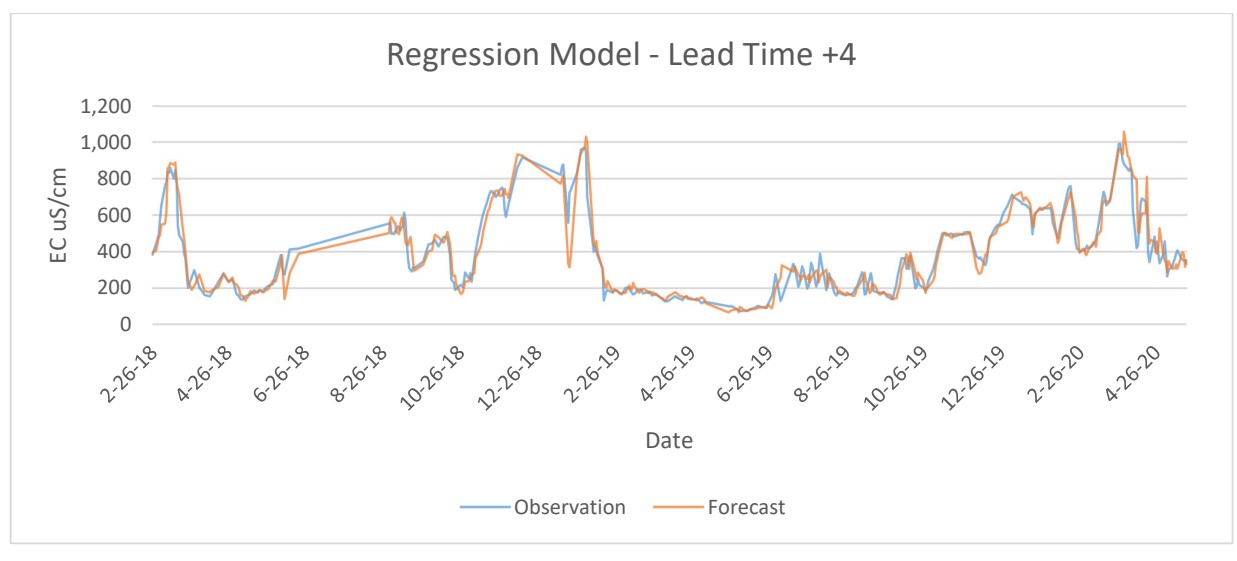

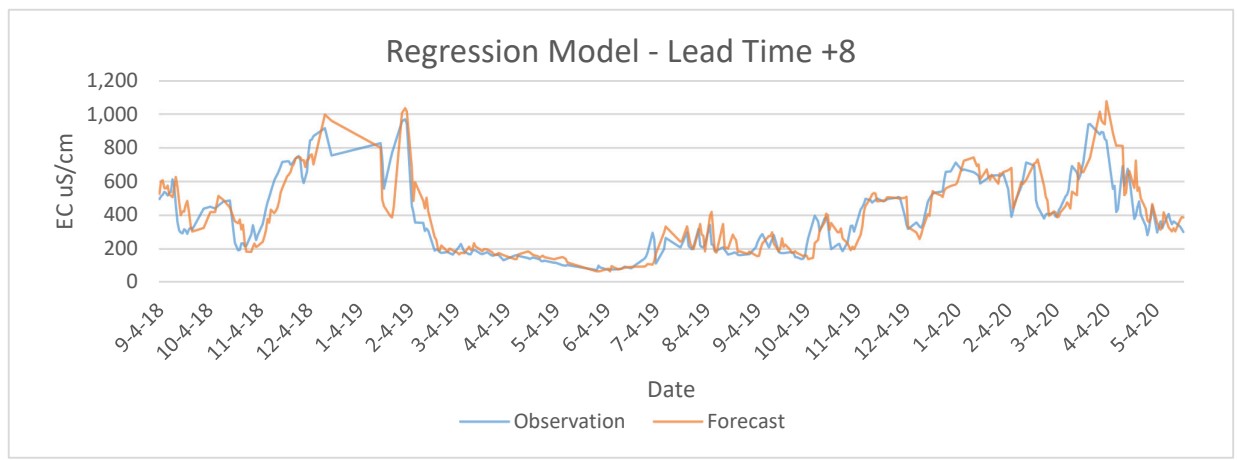

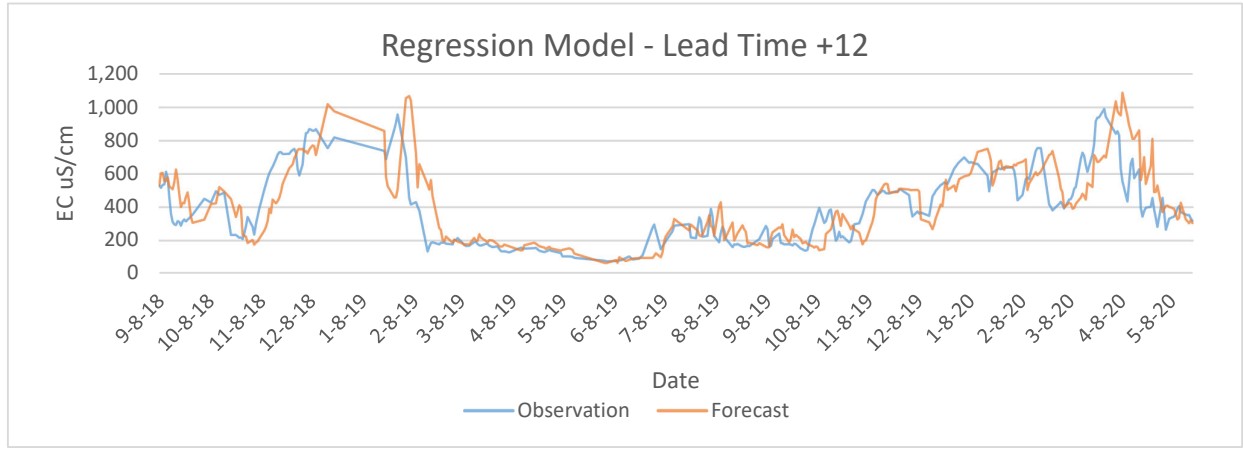

**Figure 11.** Comparison of Regression model forecasts and observations of EC at various lead times for the period between 22 February 2018 and 22 May 2020.

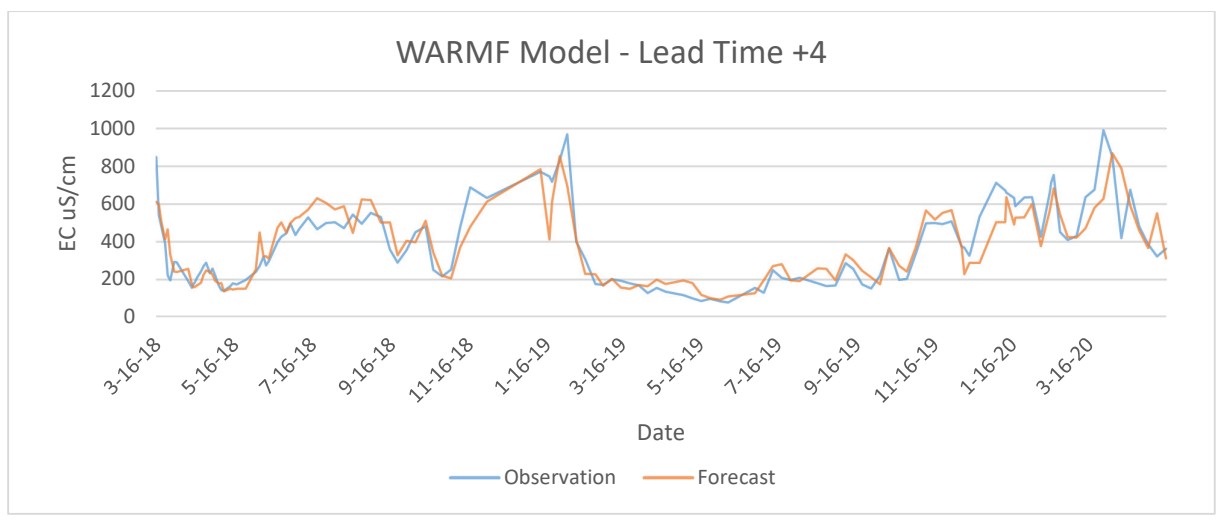

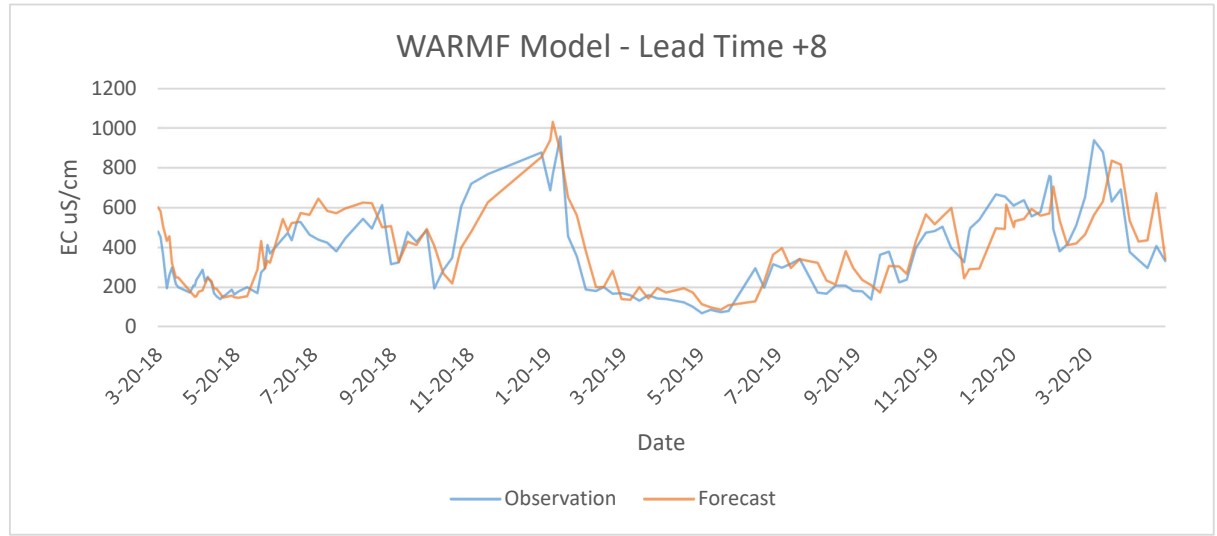

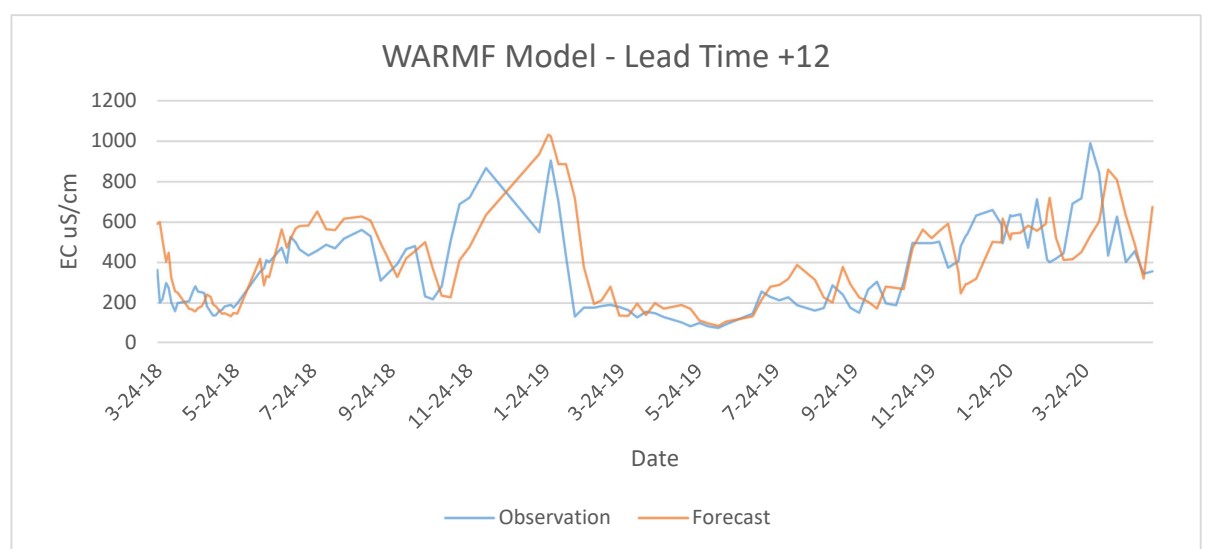

**Figure 12.** Comparison of WARMF model forecasts and observations of EC at various lead times for the period between 22 February 2018 and 22 May 2020.

## 6. Statistical Analyses

The prior analysis focused on comparisons between model EC forecasts for the regression and WARMF models and how the differences between model predictions and observations change over time for forecast lead times ranging from Δ Day + 0 to Δ Day + 14. Correspondence between model EC forecasts and EC observations exhibited significant variability. In general, as expected, the differences between model predictions of EC and observations increase with forecast lead time. Consequently, the question arises up to what lead time can the model forecasts of EC be considered reasonably reliable. In this section, a statistical approach to comparing the means of the observations and model EC forecasts is described.

The application of statistical testing methods for comparing the two models requires that careful consideration be given to the underlying assumptions made in the analysis. A preliminary decision is what statistical property should be tested. For the statistical analysis, a comparison of observation and forecast means was selected following the prior analysis based on the fact that mean salinity load, the product of the mean concentration (EC) and the mean flow, is the parameter of primary interest.

In general, most environmental data do not follow a normal distribution, as will be demonstrated for the observed EC monitoring data presented in this study. This fact has important impacts on the statistical tests that can be employed to test the equivalence of the observation and forecast EC mean values. The classical t-test statistic assumes the data are normally distributed. If they are not normally distributed, it might be possible to transform the data (e.g., using logarithmic transformations) so that, when plotted, they appear normally distributed. Such transformations can sometimes complicate the interpretation of the results. Non-parametric methods, that do not assume the data are normally distributed, are tests on the median values of the sampled data and therefore are not appropriate for this study. Another approach is the use of a permutation test. This method employs large numbers of stochastically generated realizations based on the underlying data to obtain a reasonably normal distribution of values. This is the statistical analysis approach chosen for this study.

The application of these methods was accomplished with the use of the R-commander software platform (R version 3.5.3). R is public domain software available under the "Great Truth" Copyright (C) 2019 The R Foundation for Statistical Computing. Additionally employed in the analysis were several R scripts developed by Practical Statistics Inc. and made available through their Applied Environmental Statistics courses. The statistical methods deployed in the analysis that follows were chosen based on their relative accessibility and the perception that these could be easily explained to program participants and interested stakeholders. Given the differences in the ways each of the models has been deployed for forecasting (one run daily and the other weekly), it was thought necessary to address these potential biases through the use of standard, well recognized methods. These included

1. Visual examination of the observed EC data and Regression and WARMF model EC forecasts at selected forecast lead times using boxplot graphical output.
2. Statistical testing of the normality of the observed EC data and model EC forecasts using the Shapiro–Wilks test at selected forecast lead times.
3. Statistical testing to determine whether the observed EC data and model EC forecasts have similar variances using the Fligner–Killeen test at selected forecast lead times.
4. Scatterplots of the output from the Regression and WARMF model forecasts data at selected forecast lead times.
5. Developing linear models using a forecast response variable and observation explanatory variable and computing the adjusted R-squared as an indicator of model goodness of fit at selected forecast lead times.

6.　Matched pair permutation testing to evaluate the whether the means of the observed EC and model forecast EC are statistically significant at the selected forecast lead times.

The results of these analyses are presented for selected lead times of Δ Day + 12 representing the late forecast period. The boxplots showing the results of the Regression (Figure 13a) and WARMF (Figure 13b) model forecast EC comparisons with the observed data EC. Boxplots are visual tools that can be used to indicate whether the data are normally distributed. If the distribution is normal, the boxplot would be divided into equal (blue) areas by the median (black line) and the data range represented by the dashed line would have equal lengths on the top and bottom of the box. As illustrated, these conditions are not met by the EC observations and EC forecasts for either model. The Shapiro–Wilks test is a statistical test used to evaluate whether data are normally distributed. Commonly, a $p$-value of less than 0.05 is considered indicative of a non-normal distribution. As shown in Figure 13, the $p$-values are considerably less than 0.05 confirming the boxplot interpretation. At forecast lead time Δ Day + 12, the boxplots in Figure 13 suggest that neither the observed EC or model forecast EC are normally distributed but have similar variances.

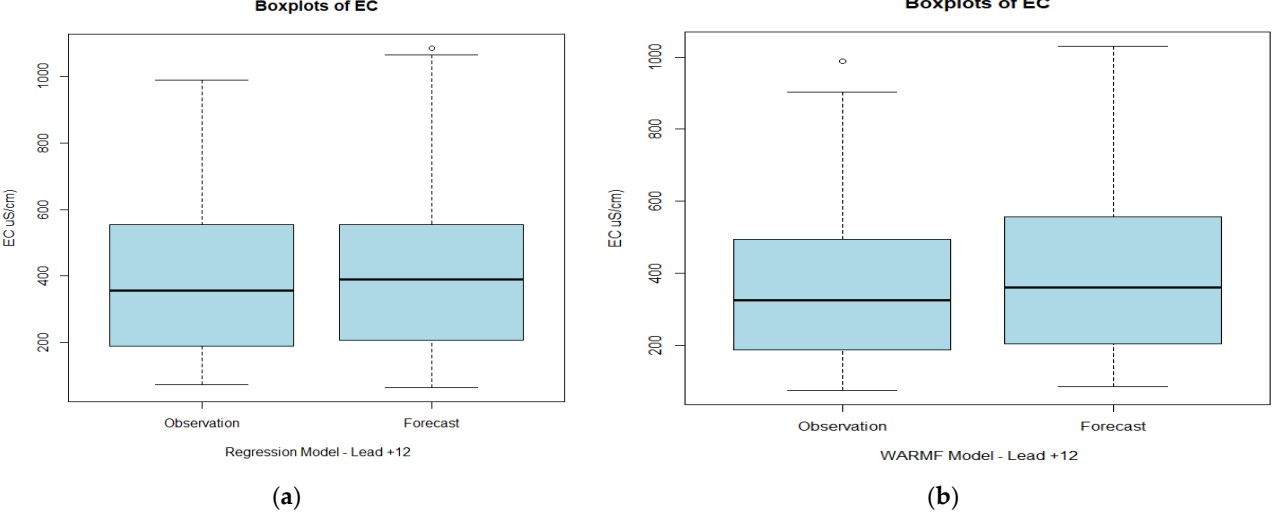

**(a)**　　　　　　　　　　　　　　　　　　　　　　　　**(b)**

**Figure 13.** (**a**,**b**). Boxplots of observed EC and forecast EC by the Regression (**a**) and WARMF (**b**) models are shown for forecast lead time Δ Day + 12. Fligner–Killeen variance $p$ values are 0.6244 and 0.2703 for the Regression and WARMF models, respectively.

Scatterplots of observed EC data and both Regression and WARMF model models EC forecasts are shown in Figure 14a,b, respectively, with their linear regression plots superimposed. The Regression model EC forecasts shows slightly less scatter around the "best fit" regression line than the WARMF model EC forecasts. However, neither model shows a high R-squared coefficient indicating poor fit.

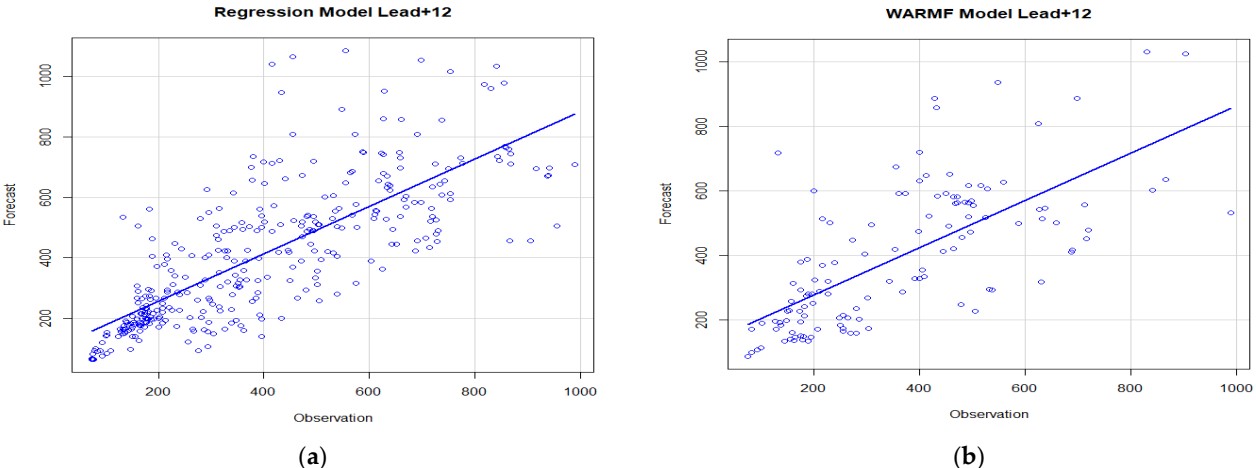

**Figure 14.** (**a**,**b**). Calculated linear regression relationship (solid blue line) for the Regression (**a**) and WARMF (**b**) models together with a scatterplot of the underlying observed EC data and model forecast EC for lead time Δ Day + 12.

Figure 15 shows the histograms and *p*-values associated with the matched pair permutation test for both Regression (15a) and WARMF (15b) model EC forecasts for forecast lead time Δ Day + 12. The results of the matched pair permutation test indicate that neither the Regression model nor WARMF model EC forecasts are good representations of the observed EC values at lead day Δ Day + 12. The Regression model EC has a *p*-value of slightly greater than 0.05 (0.1021) while the WARMF model EC has a *p*-value is slightly less than 0.05 (0.0283).

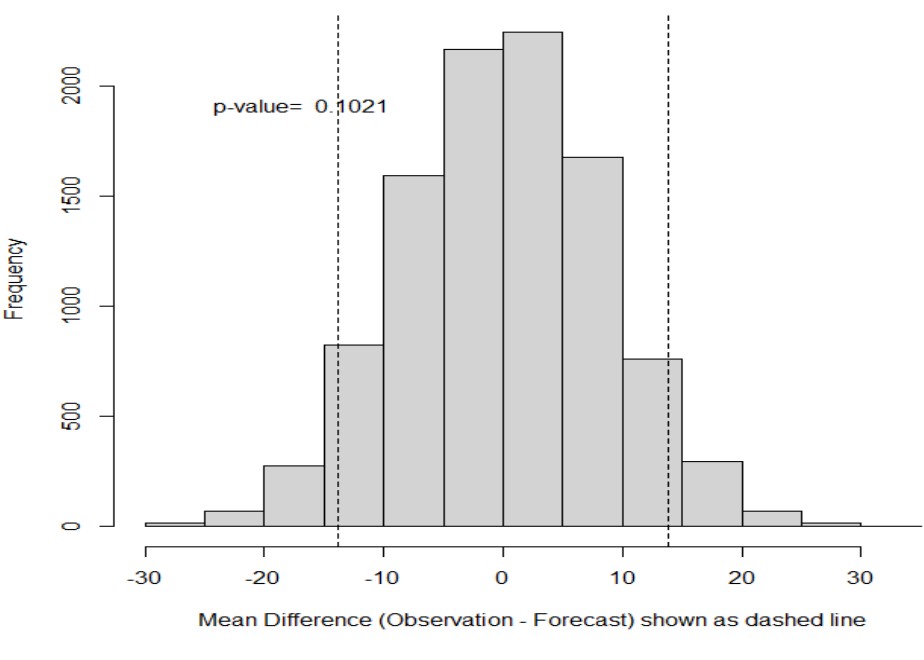

(**a**)

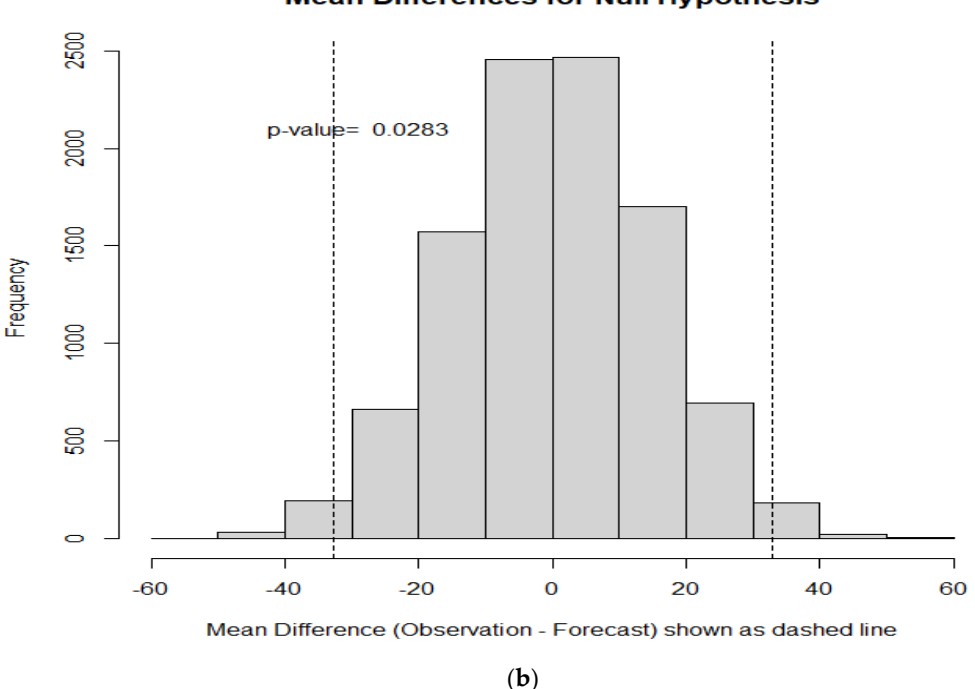

(**b**)

**Figure 15.** (**a**,**b**). Histograms of the mean differences between observed EC and model forecast EC for the Regression (**a**) and WARMF (**b**) models for model forecast lead time Δ Day + 12.

In addition to the selected lead times presented above, adjusted R-squared and matched pair permutation tests were computed for EC predictions from both Regression and WARMF models for all EC forecast lead times from Δ Day + 0 to Δ Day + 14. Figure 15a,b shows the adjusted R-squared values for both models. As illustrated, the Regression model has higher adjusted R-squared values than the WARMF model throughout the forecast period indicating a better goodness of fit. However, it is also worth noting that the adjusted R-squared values for both models decline progressively over the forecast period indicating a declining goodness of fit at longer lead times.

The results of the matched pair permutation tests comparing the mean of the observed EC and forecast EC for both regression and WARMF models are shown in Figure 16.

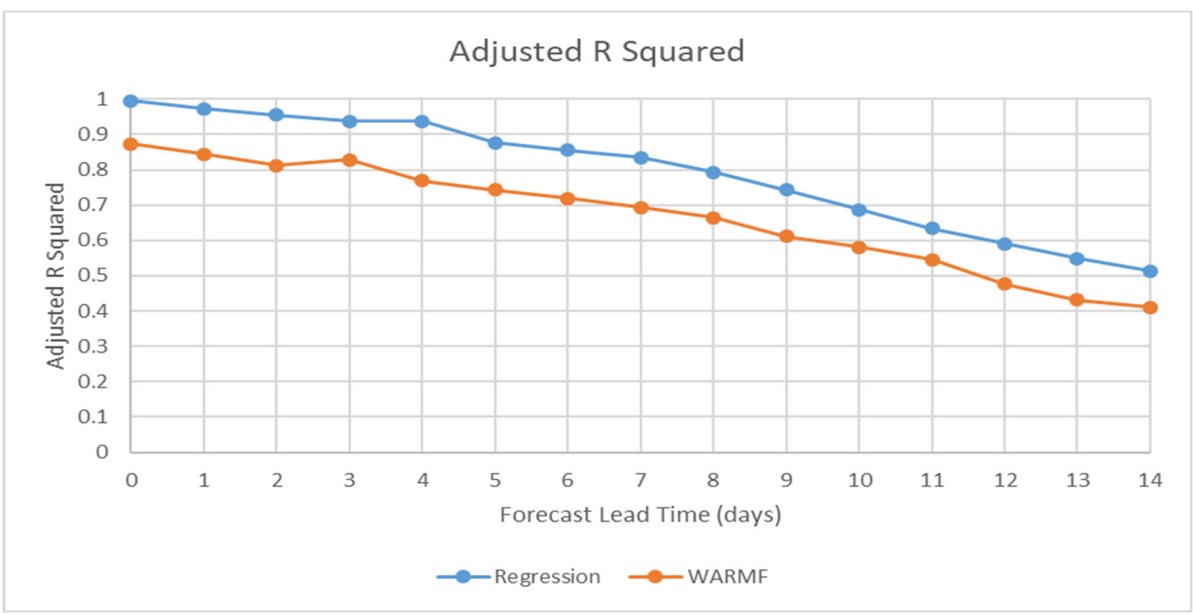

**Figure 16.** Adjusted R-squared values for the Regression and WARMF models for all EC forecast lead times.

The results of the statistical analyses are summarized as follows:

- Visual analysis and statistical tests indicate that although both observed EC and model forecast EC are not normally distributed their variance are sufficiently similar to validate the use of the matched pair permutation test to test whether the mean of the EC observations and model EC forecasts are statistically similar.
- The Regression model has consistently higher adjusted R-squared values than the WARMF model at all lead times indicating it has a relatively better goodness of fit.
- The matched pair permutation testing suggests that both models can make reasonably good EC forecasts out to approximately 7 days.

*Discussion of Model Evaluations*

　　Qualitative and quantitative comparisons of the performance of the WARMF and Regression models for forecasting EC at the compliance monitoring station at Vernalis were made to assess the utility of both models. The simple evaluation of the Regression and WARMF forecasting models comparing the differences between the observed salinity and the model-based forecasts of EC at the Vernalis compliance monitoring station between 22 February 2018 and 22 May 2020 suggested that the Regression model EC forecasts were generally closer to the overall mean of the observations than the WARMF model EC forecasts (previously shown in Figure 5). Although there were a total of 820 EC observations made at the Vernalis monitoring station fewer forecasts were made due to personnel availability and occasionally data validity issues. The WARMF model EC forecasts were made on the Monday of each week owing to the greater amount of time required to assemble model time series input data and complete each forecast and associated personnel constraints—hence forecasting frequency was roughly three times higher in the case of the Regression model (previously shown in Table 2).

　　The results of the model performance comparison as was shown in Figure 7, the Regression model provides EC forecasts with mean differences of less than or equal to 5 µS/cm for the first 7 days ($\Delta$ Day + 0 to $\Delta$ Day + 6). Alternately, the WARMF model provides EC forecasts with mean differences of less than or equal to 5 µS/cm for only 5 days ($\Delta$ Day + 0 to $\Delta$ Day + 4). Based on these measures of performance, the Regression model provides EC forecasts with reduced error relative to the WARMF model for the period from $\Delta$ Day + 4 to $\Delta$ Day + 6.

Forecast EC standard deviation, a measure of the dispersion of the EC forecasts or EC forecast differences around the mean EC value, showed that Regression model EC forecasts closely approximated of the EC observations at all lead times. The standard deviations of the WARMF model EC forecasts were consistently less than standard deviations of the EC observations until lead time day + 8. The standard deviation of forecast EC differences steadily increased with forecast lead time for both models with the WARMF model EC forecasts exhibiting greater values of standard deviation than the Regression model throughout the forecast period (previously shown in Figure 2).

To examine the effect where individual model bias affected the mean of differences between the observed EC and the model forecasted EC, EC forecast values that were higher than the measured EC were examined separately from those for which the EC forecast values were lower than the corresponding EC observations. Figures 8 and 9 showed comparisons of the positive and negative bias EC results for the Regression and WARMF models, respectively. For the positive bias differences in EC, the Regression model had smaller differences at all lead times than the WARMF model. For the negative bias differences in EC, the Regression model had smaller negative mean differences than the WARMF model. For both the positive and negative bias forecast mean differences in EC, the Regression model performed better than the WARMF model for lead times from Δ Day + 0 to Δ Day + 10. From Δ Day + 12 to Δ Day + 14, the performance of both model EC forecasts was approximately the same.

Visual inspection of the forecast EC time series results did not reveal any particular seasonal influence on the results. The RMSE between the observed EC data and model EC forecasts was also calculated as a function of forecast EC lead time. These results revealed that RMSE increased with EC forecast lead time indicating a decrease in the reliability of model forecasts. The Regression model showed consistently lower RMSE values compared to the WARMF model. The California Nevada River Forecast Center has typically run its published forecasts out only 10 days. As previously discussed, fourteen days has been considered by technical analysts associated with the real-time salinity management program to be a minimum period that would reasonably allow agricultural and wetland managers time to make adjustments to salt load export to the SJR.

Visual analysis and statistical tests suggested that neither the observed EC data or the model EC forecasts were normally distributed whereas the variances were sufficiently similar to validate the use of the matched pair permutation test, used to test whether the mean of the observed EC and model EC forecasts are statistically similar. The Regression model showed better goodness of fit relative to the WARMF model (Figure 16) as assessed by the R-squared coefficient The matched pair permutation tests indicated that both Regression and WARMF models provided reasonable forecasts extending out to approximately 7 days—not quite long enough to satisfy the goal of 14 days suggested for agricultural and wetlands stakeholder operations.

The prior analyses were based on using the full data set of all available daily observation EC data–model forecast EC paired values for both the Regression and WARMF models. However, the Regression model EC forecasts were made approximately three times more frequently than the WARMF model EC forecasts over the past 2 years (Table 3). Comparisons of the concurrent day EC forecast results with those made with the full data set suggested that the results of the analysis were similar. For both cases, the WARMF model EC forecasts were consistently lower than those the Regression model and also lower than the observed data (comparing Figure 1 with Figure 17). The standard deviations of differences between forecasts and observations for the WARMF model EC forecasts for both the full and concurrent data sets were greater than those for the Regression model at all forecast lead times.

In general, the Regression model performed better than the WARMF model for forecasting EC for up to one week into the future.

### 7. Case Study: Forecasts of EC Exceedances during Spring 2021

During February 2021, an opportunity arose to compare the forecasting capability of both models in real-time during a time period where the trend in the 30 day running average EC at two of the three SJR compliance monitoring stations suggested potential future exceedance of EC objectives. California is in the second year of a severe drought and water shortages in the State's reservoirs have resulted in severe curtailment of surface deliveries to some farmers. Federal contractors with junior water rights in the SJR Basin, south of the Delta, may receive no surface water deliveries at all during the 2021 irrigation season. The central premise of the real-time salinity management program remains that coordinated actions on the part of stakeholders can optimize the use of SJR assimilative capacity preventing violations of water quality objectives.

The real-time water quality management program was initiated during a time when Vernalis was the only compliance monitoring station for salinity on the SJR. During 2020, two additional water quality stations were added for salinity management in the lower SJR—Reach 83. This action, that was subsequently introduced as an amendment to the Basin Water Quality Control Plan, ostensibly places limits on the degradation of water quality (EC) of riparian diversions into the Patterson and West Stanislaus Irrigation Districts. Although it is unclear what enforcement actions might follow non-compliance with the new 1550 µS/cm salinity objective for Reach 83, the current WARMF model and the USBR's Regression model were extended to supply 14 day forecasts of EC and salt load assimilative capacity at these stations. The basin Plan amendment provided some compliance relief for various sequences of wet, dry and critically dry years where the 30 day running average EC limit was raised using a weighting schema. Unfortunately, the formula does not provide any means to avoid the EC objective for the current water year.

The USBR's obligation under a Management Agency Agreement (MAA) signed with the CRWQCB (the State regulator) is to meet the 30 day rolling average EC objectives at the Vernalis, Crows Landing and Maze Road Bridge, the current compliance monitoring sites for EC. These objectives are ostensibly to provide suitable water quality for riparian agricultural diversions along the mainstem of the SJR and in the Delta. The premise was that stakeholders would help to sustain water quality improvements in the SJR with the help of the USBR-funded cyberinfrastructure by scheduling drainage salt loads from west-side sources to coincide with dilution flows generated from east-side sources so as not to exceed the salt load assimilative capacity of the SJR, estimated at each of these stations.

In late February 2021, as watershed inflow to the SJR subsided after a series of rainfall events, both the WARMF and Regression forecasting models suggested a slowly increasing trend in the daily and 30 day running average EC (Figures 17 and 18) that might exceed the various compliance monitoring station EC objectives at Crows Landing and Maze Road (EC 30 day running average objectives of 1550 µS/cm) and at Vernalis which was transitioning from the winter 30 day running average EC objective of 1000 uS/cm to the irrigation season objective of 700 µS/cm. Note that the irrigation season objective applies after April 30 (when 30 days have elapsed). The weekly WARMF model forecast (green background) suggested on 2/22/21 that the 30 day running average EC threshold of 1550 µS/cm at Crows Landing could be exceeded on 6 March 2021 (Figure 18a) whereas the Vernalis site still showed salt load assimilative capacity (Figure 17a). The USBR had been making regular adjustments of New Melones reservoir releases to maintain compliance with EC objectives at Vernalis as required under the MAA. The Regression model (blue background) that was run on the same Monday February 22 (Figure 18b) suggested an occurrence of the same exceedance event although the date of the exceedance was predicted one day earlier. In order to lower the 30 day running average EC at Crows Landing, west-side return flows upstream of Crows Landing would need to fall below the 1550 µS/cm criterion.

WARMF and Regression model forecasts made on April 26 were much closer in their predictions (Figure 18c,d) and neither suggested that 30 day running average would drop below the zero line—indicating continuing exceedance and lack of SJR SLAC (Figure 18e,f). The forecasts made by the models on 6/1/21 show that the daily mean EC dropped below the objective on 5/19/21 and continued to drive the 30 day rolling average downward until it dropped below the 1550 μS/cm objective and transitioned into positive territory on 5/28/2021 (Figure 19a,b).

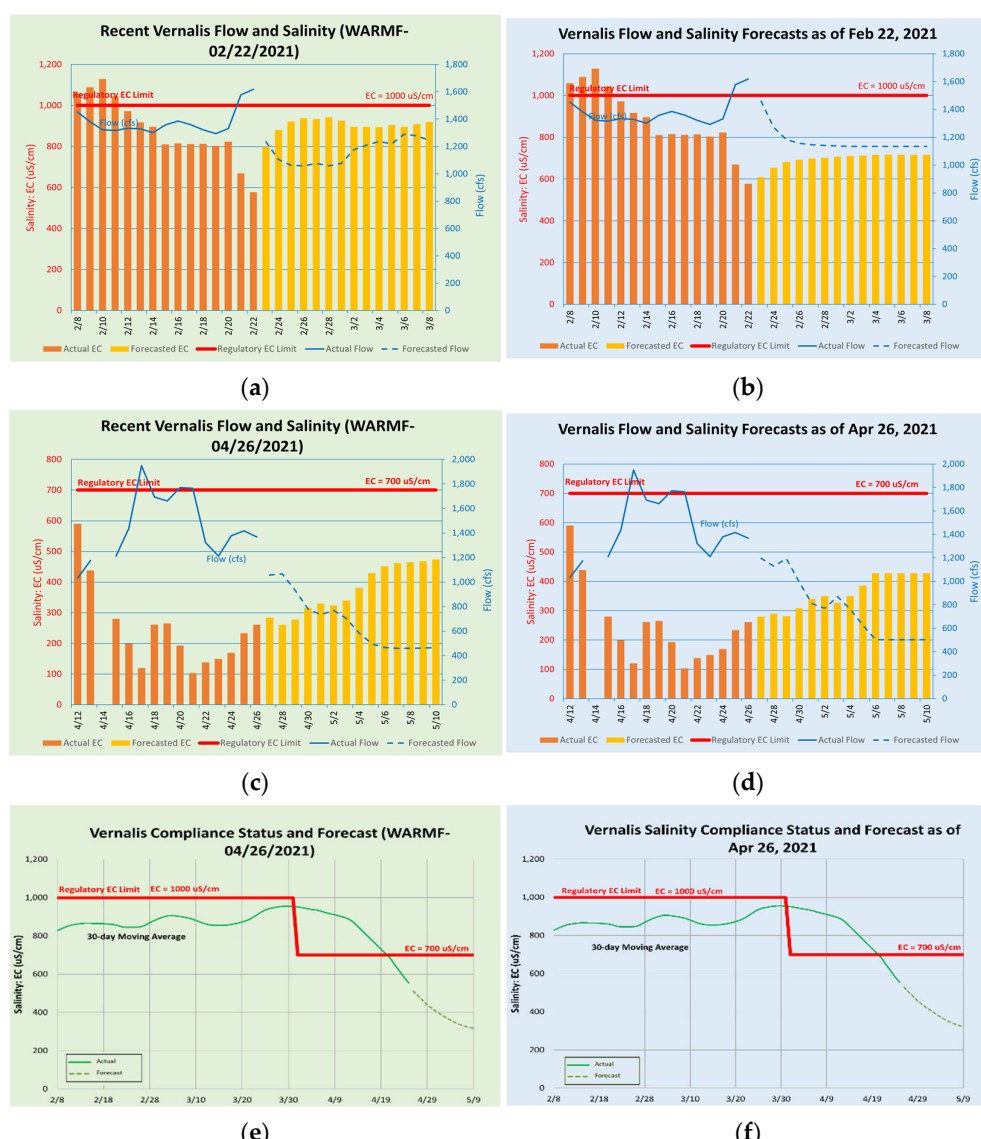

(**a**)                 (**b**)

(**c**)                 (**d**)

(**e**)                 (**f**)

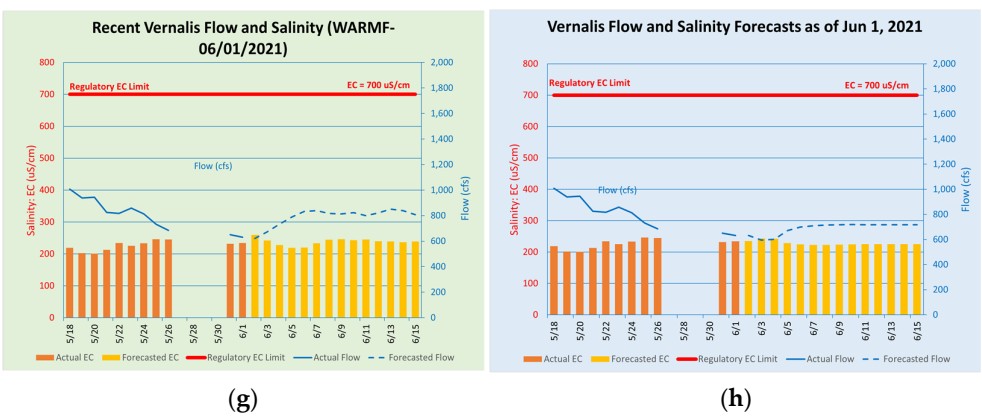

(**g**)　　　　　　　　　　　　　　　　(**h**)

**Figure 17.** Comparison of daily WARMF and Regression model forecasts for EC at the Crows Landing compliance monitoring station on 2/22/21 (**a,b**); 4/26/21 (**c,d,e,f**); and 6/01/21 (**g,h**). Graphs (**e,f**) show the 30-day running average EC forecast on 4/26/21 relative to the the 30-day running average EC compliance objective. Conversion of flow in cfs to m³/sec: 100 cfs = 2.83 m³/sec.

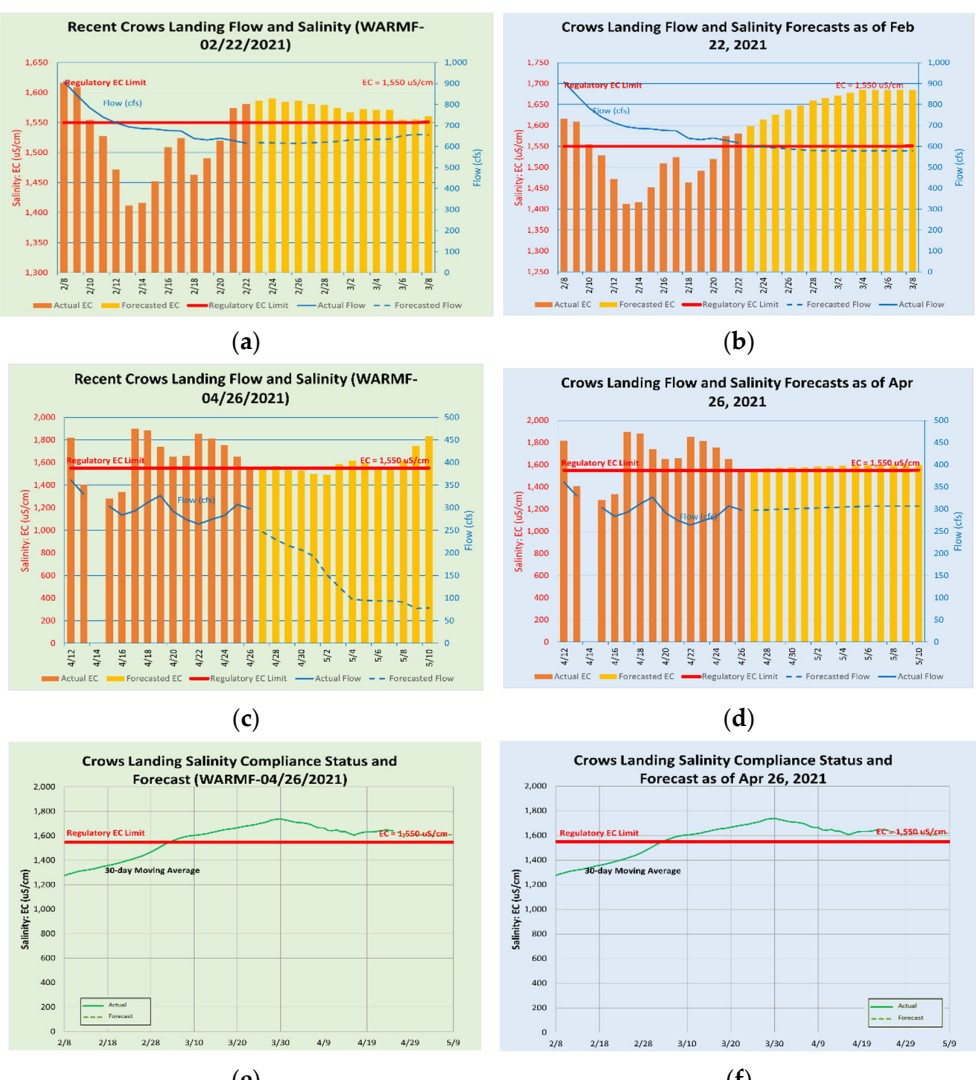

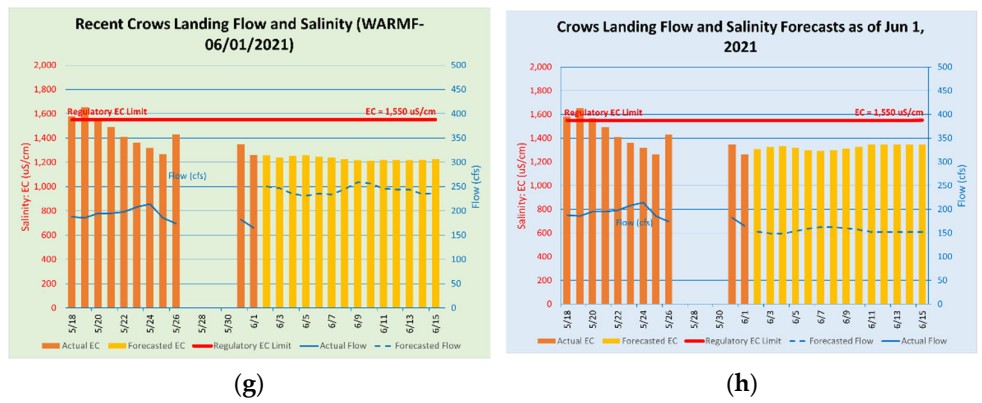

(**g**) (**h**)

**Figure 18.** Comparison of daily WARMF and Regression model forecasts for EC at the Crows Landing compliance monitoring station on 2/22/21 (**a,b**); 4/26/21 (**c,d,e,f**); and 6/01/21 (**g,h**). Graphs (**e,f**) show the 30-day running average EC forecast on 4/26/21 relative to the the 30-day running average EC compliance objective. Conversion of flow in cfs to m3/sec: 100 cfs = 2.83 m³/sec.

This transition is also shown in Figure 19. Figure 19c, which was produced on 6/1/2021, correctly predicted the transition to positive salt load assimilative capacity on 5/26/2021. This plot also shows the proportion of the salt load contributed by the combination of Mud and Salt Slough relative to the total salt load measured at the Crows Landing compliance monitoring station. At this time of year, the majority of the salt load in these Sloughs is seasonal wetland drainage, which typically has an EC in excess of 1500 µS/cm.

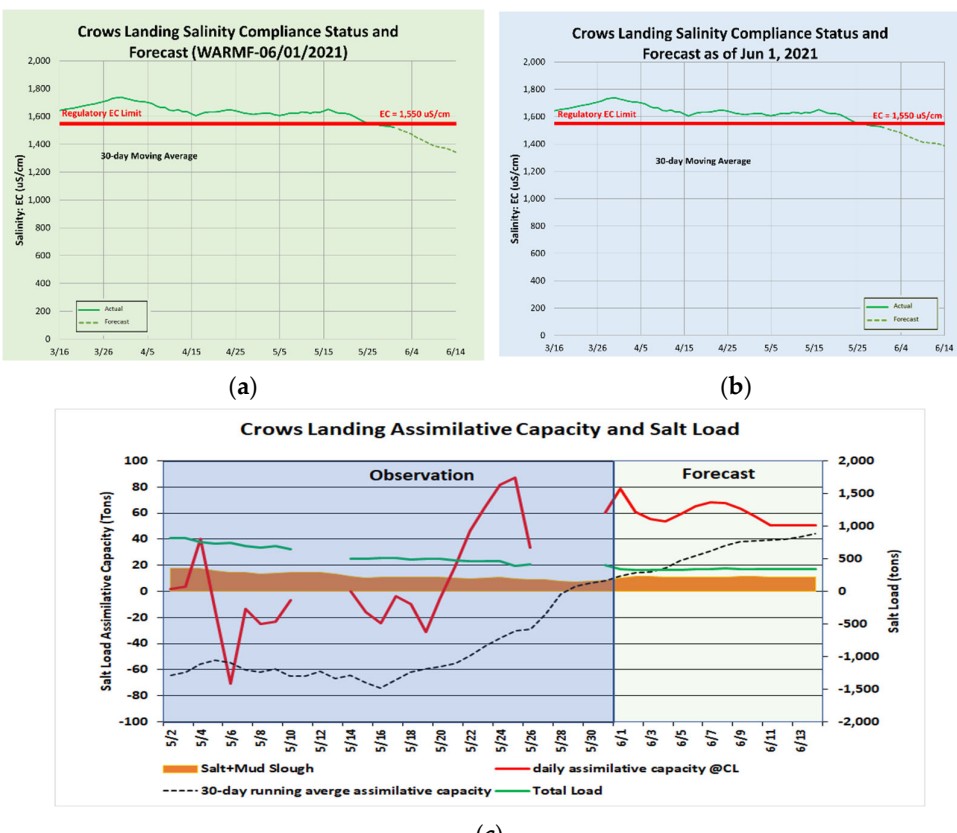

**Figure 19.** Comparison of daily WARMF and Regression model forecasts for EC at the Crows Landing compliance monitoring station on 6/1/21. Figures (**a,b**)show the 30 day running average EC and

forecast for 6/1/21. Figure (**c**) shows the SLAC at the Crows landing station. By early May wetland drainage no longer dominates Mud and Salt Sloughs and daily SLAC in the river increases. The 30 day running average SLAC crosses the zero line around 28 May 2021. Breaks in the plot are the result of temporary EC sensor malfunction at the Crows Landing station. Conversion of flow in cfs to m³/s: 100 cfs = 2.83 m³/s.

## 8. Stakeholder Response and Coordination

As previously noted, this event has provided the USBR with an opportunity to demonstrate the agency's commitment to its obligations under the MAA, reminded stakeholders of their role in the real-time program and exposed deficiencies in real-time response to periods of water quality exceedance. During the second week of February, when it became clear through the use of the forecast models that the salinity at both Vernalis and Crows Landing stations was trending towards potential exceedance of the 30 day running average EC stakeholders were notified directly. The likely date of exceedance was estimated to be March 5 from WARMF and Regression model forecasts made on February 23, 2021. In order to provide stakeholders adequate time to perform remedial actions, we decided to directly engage with stakeholders in the SJRB rather than rely on the USBR's normal weekly posting of flow, EC and 30 day running average EC at the three compliance monitoring stations. Communication with stakeholders was primarily by e-mail to east- and west-side agricultural stakeholder coalitions, directly impacted water district, and representatives of the private, state and federal wetland entities. the San Joaquin Valley Drainage Authority, Grassland Water District, Los Banos Wildlife Management Area, Patterson and West Stanislaus Irrigation Districts, on the east-side Modesto and Turlock irrigation districts and the East SJR Water Quality Coalition. A similar e-mail was sent to the Regional Water Quality Control Board, the basin regulator, that has the power to set fines for water quality objective exceedances.

In retrospect, the timing of the stakeholder outreach was timely and prescient. Although anticipated, programmatic fish migration flows from east-side reservoirs, that started in mid-April, were able to drive down the EC at Vernalis below the 700 uS/cm limit that came into effect on April 30. The Merced River is the only tributary to the SJR upstream of Crows landing and supplemental flows for fish migration were insufficient to prevent the EC at Crows landing from exceeding objectives. During the period of exceedance at the Crows Landing compliance monitoring, there were opportunities to address the excess salt loading to the SJR. During the initial period of exceedance, raising the board elevation at the San Luis Drain outlet (a previously used conveyance facility that carries only subsurface agricultural drainage) and storing drainage return flows in the drain for later release would have reduced salt loading by 100 tons (91 tonnes) per day and eliminated the deficit in SLAC. Most drainage return flows into the drain are from seepage from adjacent agricultural land and wetland and with an average salinity of 2500 uS/cm. However, after the first days of exceedance, the daily EC remained elevated above 1550 μS/cm and the 30 day running average SLAC deficit climbed to a steady state load of approximately negative 1000 tons (907 tonnes) per day.

The exceedance of the Crows Landing EC objective occurred during the wetland drawdown period when the Grassland Water District and adjacent State and Federal refuges are draining ponded surface water to allow germination of swamp timothy, smartweed and water grass food crops that serve overwintering waterfowl. Since the timing of this drawdown is critical for swamp timothy production and the waterfowl that prefer this food source, asking wetlands to curtail drawdown during this period was viewed as unrealistic by wetland resource managers.

Procurement of additional dilution flow from the Merced Irrigation District was also unrealistic given the prevailing drought conditions and anticipated water shortages during the summer of 2021. In addition, some entity would have had to foot the bill for procurement of any additional supply if supply were available.

The Regional Board has taken a "wait and see" approach to this first test of the real-time water quality management system and the newly promulgated upstream EC objectives at Crows Landing and Maze Road compliance monitoring stations. There has been no discussion of fines or allocation of penalties across subareas contributing salt load to the SJR from the Regional Board. There is also the fact to consider that riparian diverters along the northwest-side subarea is the river reach that the upstream objective was promulgated to protect. Fining stakeholders who are being harmed by the elevated EC along this reach of the SJR would be problematic.

At the time of writing, the severe drought conditions in the basin have reduced forecasted flow for June 10, 2021 at the Crows landing compliance monitoring station to under 100 cfs and daily EC is once again over the 1550 threshold EC. The 30 day running average EC is climbing once again and may remain above the objective for the remainder of the irrigation season while drought mitigation actions are in force.

## 9. Summary and Conclusions

Real-time salinity management is a stakeholder- and water agency-sanctioned program that helps to maximize allowable salt export from the agriculture-dominated SJR Basin. The essential components of the current program that are now in place include the establishment of telemetered sensor networks, a web-based information system for sharing data, a basin-scale salt load assimilative capacity forecasting model and institutional entities tasked with performing weekly forecasts of river SLAC and using these forecasts to improve scheduling of west-side drainage salt load export and the dilution provided by east-side reservoir releases. Two modeling approaches were developed simultaneously, in part to see if a higher level of automation could be introduced in developing SLAC forecasts and if the frequency of these forecasts could be moved from weekly using the WARMF numerical simulation model to a simpler flow-based regression modeling approach run daily. The Regression model relies on a comprehensive statistical analysis of the relationship between flow and salt concentration at three compliance monitoring sites. The WARMF watershed water quality simulation model provided the conventional SLAC forecasting approach. The model is data driven and although model data acquisition is almost fully automated, there is still a need for user involvement for simulation times that may take an hour or more. The results from both models are migrated manually to Excel spreadsheets that are used to produce graphics that are posted to the web daily in the case of the Regression model and weekly for the WARMF model.

The first part of this paper has provided a comprehensive analysis of the model results when used to make 14 day EC forecasts (daily and 30 day running average EC) and an estimate of 14 day river SLAC. Analysis of the results from both model-based forecasting approaches over a period of five years shows that the regression-based forecasting model, run daily Monday to Friday each week, provided marginally better performance. However, the regression-based forecasting model assumes the same general relationship between flow and salinity which breaks down during extreme weather events such as droughts when water allocation cutbacks among stakeholders are not evenly distributed across the basin. A recent test case was used to demonstrate the potential utility of both models in dealing with an exceedance event at the Crows Landing compliance monitoring station. This year is providing an opportunity to test the robustness and reliability of the flow-EC relationship that the regression model relies upon since contract water delivery to USBR contractors is scaled back unequally during times of shortage in association with District water rights. The major lesson learned from the project to date is that a dual modeling approach of using a simple Regression model for daily automated forecasting with weekly simulation model runs using the WARMF model appears to be a good compromise at present that provides sufficient frequency of forecasts to allow stakeholders to make timely decisions (Regression model) while using stakeholder data to eliminate model inconsistencies during periods of unusual or extreme basin hydrology. The use of the WARMF model in this dual modeling approach provides modelers with a tool to more

fully understand the current state of the system and to investigate unusual occurrences in basin hydrology and water quality that are only possible with a mechanistic model like the WARMF model.

In the future, it would be desirable that the Regression and WARMF models are both run daily which would eliminate some of the model comparison questions that were addressed in this study. Further automation of WARMF model data pre-processing steps could be combined with similarly automated real-time data quality assurance routines—perhaps enhanced with machine learning procedures to eliminate data gaps, remove sensor drift and data spikes to improve model performance. The lack of a robust and customizable, public domain real-time data quality assurance software tool remains the biggest remaining impediment to water quality forecasting capabilities and if addressed could enhance stakeholder confidence in this instance of model-based environmental decision support.

**Author Contributions:** Conceptualization, N.W.T.Q.; methodology, N.W.T.Q., M.K.T. and T.J.L.; software, T.J.L.; statistical analysis, M.K.T.; data curation, M.K.T., T.J.L. and N.W.T.Q.; visualization, M.K.T. and N.W.T.Q.; writing—original draft preparation, N.W.T.Q.; writing—review and editing, N.W.T.Q. and M.K.T.; response to reviewer comments, N.W.T.Q. All authors have read and agreed to the published version of the manuscript.

**Funding:** This research received no external funding.

**Institutional Review Board Statement:** Not applicable.

**Informed Consent Statement:** Not applicable.

**Data availability statement:** No new data, models, or code were generated or used during the study—this study has com-piled publicly accessible data and existing model output to create a unique contribution to the literature. Daily model forecasts of flow and EC at the three compliance monitoring sites mentioned in the paper produced using the Regression model are available on the USBR's web portal at http://www.usbr.gov/ptms/ (accessed on 20 September 2021). Weekly WARMF model forecasts may be substituted for the Regression model forecasts when SJR flow data is unavailable or unreliable. These forecasts are posted on the same web portal.

**Acknowledgments:** The lead author wishes to recognize individual contributions made to the manuscript by the co-authors within the USBR. Much of the statistical analysis in the paper was drawn from an internal agency report provided by Michael Tansey. James Lu developed the EC forecasting algorithm for the Regression model and is responsible for daily web posting of Regression model-based forecasts of EC at the three SJR compliance monitoring sites. WARMF model forecasts are made weekly by Jun Wang and the lead author and are posted by Jun Wang on the same USBR web portal. Recent results from both models were used in the case study presented. The lead author also wishes to acknowledge the support of the San Joaquin Valley Drainage Authority and member water districts including Grasslands Water District, Patterson and West Stanislaus Irrigation Districts that have helped to advance the concept of real-time water quality management. Also, the California Regional Water Quality Control Board, the State regulator, for encouraging this new approach to salt management in the Basin that maximizes the beneficial use of the SJR. Much of the success of the Program is a result of generous State of California Department of Water Resources funding under Proposition 84 which has allowed upgrading and ongoing support of the various flow and EC sensor networks. The USBR has provided essential ongoing support since the inception of the Program.

**Conflicts of Interest:** The authors declare no conflict of interest.

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
