# Peer review of "Comparison of Deterministic and Statistical Models for Water Quality Compliance Forecasting in the San Joaquin River Basin, California"

_water, doi:10.3390/w13192661_

Round 1

Reviewer 1 Report

The author has reported the development and application of two salinity forecasting models in the San Joaquin River Basin in California, which serve as decision support tools for the federally and state mandated management of salt in the waterways of the San Joaquin River. While the paper is generally well written, and does a remarkable job of laying the background of the salinity issue in the California Central Valley, there are some minor concerns that need to be addressed:

  1. The abstract and introduction section must include one or two sentences discussing why the two modeling approaches were chosen and what the intended application and benefits of each are. The motivations for the physics-based and statistical approaches in the first and last paragraphs of Section 3.1 and the first paragraph in Section 3.2 should be compressed and presented in the abstract and introduction. Additionally, as a significant portion of the paper focusses on a thorough inter-model performance comparison, it would be useful to state this as one of the objectives of the paper. This way, the paper would also serve as a guide to model developers on how to compare disparate models of complex water systems. As written, these sections end somewhat abruptly.
  2. As written, the manuscript requires readers to have an intimate familiarity with the system being discussed. I would urge the author to provide a gentler introduction into the complex issues of salinity management here for a wider global audience. Please use the narrative in Sections 2 and 3 and Figure 1 with appropriate legends to lay out the water quality stressors, and management drivers in the system.
  3. Throughout the paper, the author has peppered in interesting tidbits of information on how local domain-expertise has served the construction of practically useful models. Such knowledge is essential to move modeling from the purely theoretical domain to practice. I would recommend that a paragraph be included in the discussion section to collect these insights and present a narrative on how local expertise should be factored into model development.
  4. I would have liked to see a brief discussion on how the assumptions in the WARMF model compare with those in CALSIM III, another widely used water management model in this region. The author's insights into how these two models would coexist in the region's management ecosystem would be very useful in understanding the challenges of decision support in a multi-model environment.
  5. The shorter divergence timescale and unbounded growth of forecast error for the WARMF model, as compared to the timeseries regression indicates that either (1) salt loading in the San Joaquin is an inherently chaotic process, or (2) the WARMF model is not able to reliably simulate the key physical processes. A discussion of these results would be very useful to guide model deployment for similar applications.
  6. Please collect Sections 4 through 7 into a new "Results" section to streamline the paper.
  7. I am not sure that Section 6.1 adds anything to the paper. A subsetting of the data based on what seems to be an arbitrary choice, i.e., concurrent day forecasts, indicates qualitatively different results from the earlier sections. I recommend dropping this section altogether, as it brings the statistically robust results of the earlier sections that the regression model outperforms the WARMF model and that the statistical model has a larger forecast horizon than the WARMF model into question. Based on the rigorous tests until this point, its clear that these results are robust, so this arbitrary subsetting of the data seems unwarranted.
  8. The resolution of some of the figures have to be improved to meet the journal requirements. Please see the relevant sticky notes in the attached pdf.
  9. Please ensure that figure references are correct.
  10. Figures 22 and 23 are extremely complicated multi-panel figures. The central ideas of compliance exceedence in certain periods is conveyed by the simpler (e) and (f) subplanels in these figures. I recommend retaining just these for clarity and increasing their size so that the font is readable, and removing the other bar charts.
  11. I recommend carefully reviewing the acronyms to see if they are really needed. There are way too many that appear in the initial third of the paper only once.
  12. Mostly minor recommendations are in sticky notes in the attached PDF.

Author Response

Please see attached file.  Authors are particularly grateful to reviewer 1 for his/her careful review and useful suggestions for improvements to the manuscript.

Reviewer 2 Report

The article titled “Comparison of Forecasting Tools for Compliance Decision Support in Support of a Real-Time Salinity Management Program in the San Joaquin River Basin, California” does a comparative study between two different forecasting tools for supporting real-time salinity management programs. The study has been done diligently, but some critical shortcomings must be addressed before the article is ready for submission.

The abstract does not stand alone, and it does not reflect the presented research. Although the method section discusses the models and modifications within the model, it does not discuss how the models were compared and why they were compared on those parameters. Therefore, the reader has to guess why these parameters were selected for comparing the models in the result and discussion section. Further, parts of the method are within the result section and make it hard to follow as a reader. The discussion section lacks relevant current work in the same line of thought and does not compare the study findings with the literature, and there is no citation in the discussion section. These sections of the paper need to be improved and reorganized for an easy read and to maintain the flow of discussions.

Specific comments,

  • The title needs to be rephrased and possibly shorter.
  • Figures 1 and 2 needs to be improved. Further, the number of figures can be reduced for a more coherent study. Some figures are hard to read when printed.
  • Figure 1 description in the text does not match with the representation in the figure.
  • The subclass should be a subsection within the regression model, and else it feels like the author is comparing multiple models instead of two distinct systems.
  • Model description can be shortened, and other methods discussed in the result need to be part of the method section.
  • A clear outline has to be presented why specific parameters were selected for comparison.
  • There are two sections 7, please check and number accordingly.
  • There is no single citation in the discussion section of the article. The results should be compared with relevant literature and present what is different in the current study.

Author Response

The authors thank  Reviewer 2 for his/her comments.  Response to comments is provided in the attached file.  A track -changes version of the manuscript has been provided for the reviewer to see changes to the original manuscript.

Reviewer 3 Report

The paper topic is very interesting however the paper must be completely revised to enhance its scientific quality.

The paper is too long: the paper should focus on the modelling approaches while leaving out to many details about the specific case study. 

It does not give important information about the key points: 

  • model characteristics
  • model parameters (how these parameters were identified?)
  • model uncertainties (what is the impact of model structure and parameter uncertainties?)
  • model forcing (what weather forecasts are used?)
  • model results (statistical indices are not adequate for assessing model performances; see for instance analogous papers concerning forecast performances at different lead times, e.g. Pelosi et al Agricultural Water Management 2016)

The paper mention ANN based model but apparently the model is not applied.

The quality of the figures is poor: they do not reflect MDPI editorial standards.

Author Response

(The authors gave the same response as above.)

Round 2

Reviewer 2 Report

The authors have addressed most of the comments. However, in the revised version pdf file figure 1 has been cropped and missing part of it on the right side. I am not sure if this was intentional or the generated pdf cropped the image. I understand the authors' points in Figure 2, still, the words within the figure are hard to read in the printed form. Though the authors mentioned the revised version sections have been renumbered, I still see two sections 7 (lines 784, 989) in the revised manuscript, which would need to be corrected. Overall, the manuscript reads well, thank you for the opportunity to review this article.

Author Response

Please see attached file.  Authors are particularly grateful to reviewer 2 for his/her careful review and useful suggestions for improvements to the manuscript.

Reviewer 3 Report

I do not fully agree with authors' response.

I suggest removing ANN section: if you do not present the results of a method you cannot present it in the method section.

The fact that you applied ANN in your  project does not justify its insertion in the methods section.

You could mention ANN o other alternative techniques in the introduction.

Figures quality is still poor.

Tables and figures do not reflect journal standards.   

Author Response

Please see attached file.  Authors are particularly grateful to reviewer 3 for his/her careful review and useful suggestions for improvements to the manuscript.
